# Visualizing cortical laminar architecture in the living human brain using next-generation ultra-high-gradient diffusion MRI
Hansol Lee [1,2,3,10], Yixin Ma[1,2,10], Kwok-Shing Chan [1,2], Eva A. Krijnen[4,5], Laleh Eskandarian[1,2], Aneri Bhatt [1,2], Julianna Gerold [1,2], Mirsad Mahmutovic[6], Oula Puonti[1,7], Xiangrui Zeng[1,2], Lucas Jacob Deden Binder[1,2], Bruce Fischl [1,2], Boris Keil[6,8,9], Gabriel Ramos-Llordén[1,2], Eric C. Klawiter[4], Hong-Hsi Lee [1,2,11] ✉ & Susie Y. Huang [1,2,11] ✉

Characterizing cortical laminar microstructure is essential for understanding the organization of the human brain. Leveraging the next-generation Connectome MRI scanner (maximum gradient strength=500mT/m, slew rate=600 T/m/s), we characterized in vivo cortical laminar cytoarchitecture and myeloarchitecture through cortical depth-dependent analyses of soma and neurite density imaging (SANDI) metrics derived from 1 mm diffusion MRI generated using a super-resolution technique. SANDI revealed distinct laminar profiles: intra-soma signal fraction peaked at ~55% cortical depth, while the intra-neurite signal fraction increased toward deeper cortical layers, consistent with known histological patterns. The visual cortex showed higher intra-soma signal fraction than the motor cortex, particularly in deeper layers. Intra-soma signal fraction correlated positively with cortical curvature in superficial layers and negatively in deeper layers, indicating layer-specific relationships between cortical microstructure and geometry. These findings demonstrate the feasibility of noninvasive mapping of laminar architecture, offering a potential in vivo surrogate for histology and enabling future studies of cortical laminar organization using high-performance gradient MRI.

The cerebral cortex exhibits diverse and highly intricate cyto- and myeloarchitectonic organization underpinning motor, sensory, cognitive, and other functions that are central to human brain function[1-6]. The size and density of neuronal and glial cell bodies, as well as the distribution of myelin, exhibit distinct profiles across cortical depths, a characteristic known as laminar organization[7,8]. Cortical laminar patterns are classically divided into three compartments: supragranular layers (I–III), the granular layer (IV),

and infragranular layers (V–VI), together forming the six-layered structure of the neocortex (I–VI)[9–11]. Each layer is characterized by distinct molecular composition, cellular architecture, and connectivity patterns. Connections within and between cortical columns are organized in layer-specific patterns that support hierarchical information processing. These structural and connectivity features collectively give rise to the functional specialization of cortical regions within distributed neural circuits[1,2,5,11]. Characterizing cyto-

[1]Athinoula A. Martinos Center for Biomedical Imaging, Department of Radiology, Massachusetts General Hospital, Charlestown, MA, USA. [2]Harvard Medical School, Boston, MA, USA. [3]Department of Biomedical Engineering, Ulsan National Institute of Science and Technology, Ulsan, South Korea. [4]Department of Neurology, Massachusetts General Hospital, Harvard Medical School, Boston, MA, USA. [5]MS Center Amsterdam, Anatomy and Neurosciences, Amsterdam Neuroscience, Amsterdam UMC, Location VUmc, Amsterdam, The Netherlands. [6]Institute of Medical Physics and Radiation Protection, TH-Mittelhessen University of Applied Sciences, Giessen, Germany. [7]Danish Research Centre for Magnetic Resonance, Centre for Functional and Diagnostic Imaging and Research, Copenhagen University Hospital-Amager and Hvidovre, Copenhagen, Denmark. [8]LOEWE Research Cluster for Advanced Medical Physics in Imaging and Therapy (ADMIT), TH-Mittelhessen University of Applied Sciences, Giessen, Germany. [9]Department of Diagnostic and Interventional Radiology, University Hospital Marburg, Philipps University of Marburg, Marburg, Germany. [10]These authors contributed equally: Hansol Lee, Yixin Ma. [11]These authors jointly supervised this work: Hong-Hsi Lee, Susie Y. Huang. ✉e-mail: hlee84@mgh.harvard.edu; susie.huang@mgh.harvard.edu

and myeloarchitectonic features in the living human brain is therefore critical for understanding how cortical microstructure supports cognition, sensory processing, and motor control, as well as how alterations in laminar organization contribute to neuropsychiatric and neurological disorders.

Detailed examination of brain tissue microstructure has traditionally relied on postmortem microscopic imaging techniques, including histological staining, electron microscopy, and immunofluorescence[12–14], which enable direct characterization of cellular components and their spatial organization. Ex vivo magnetic resonance imaging (MRI) provides a powerful complementary approach, enabling imaging of larger regions of postmortem brain specimens compared with traditional histological techniques and allowing whole-brain imaging at resolutions approaching 100 μm without tissue sectioning[15–18]. Ex vivo MRI reveals detailed neuroanatomical features, such as cortical lamination[19,20], hippocampal subfield architecture[21], and white matter organization[22], as well as pathological protein deposition, such as amyloid plaque distribution[23] and vascular abnormalities[24] associated with neurodegenerative diseases. While these approaches have greatly contributed to our understanding of brain anatomy and pathology, their inability to capture physiological processes or functional connectivity, together with artifacts introduced by tissue fixation and processing[25], limits their ability to study the dynamic properties of the living brain. The development and advancement of in vivo neuroimaging techniques[26–30], driven by innovations in MRI technology, are therefore essential for bridging the gap between postmortem histology and translational research, enabling unprecedented investigation of microstructural features, physiological processes, and functional connectivity in the living human brain. Anatomical MRI is effective for identifying macroscopic features, such as large-scale brain organization and structural abnormalities, such as atrophy or lesions[31–33]. Diffusion MRI (dMRI) complements these structural insights by probing tissue features at the cellular level, enabling the detection of subtle changes in tissue microarchitecture, including alterations in axonal integrity, synaptic density, and cellular composition[34–36].

Over the decades, substantial progress has been made toward noninvasive visualization of cortical laminar architecture in the living human brain using advanced MRI methods[11,37]. These efforts utilize different contrast mechanisms ($T_1$, $T_2$, $T_2^*$, phase, susceptibility, magnetization transfer, and dMRI) to study cortical gray matter organization at the laminar level[38–42]. Among these methods, $T_1$-weighted approaches have been the most extensively studied and have proven to be robust and practical. Early studies showed that myelination shortens $T_1$ relaxation times and that $T_1$-weighted images contain laminar signatures reflecting the underlying myeloarchitecture, revealing six $T_1$-defined components that correspond to histological layers[43–45]. Subsequent studies extended this framework using quantitative $T_1$ inversion-recovery MRI protocols to enable whole-brain laminar mapping while addressing partial volume effects[46,47], modeling cortical laminar connectivity[48–50], and applying laminar imaging in clinical populations, such as detecting dyslamination in epilepsy[51]. Although $T_1$-derived laminar patterns have recently been shown to correlate with cytoarchitectonic regions[52], $T_1$ relaxation primarily reflects myeloarchitecture rather than cytoarchitecture and therefore provides an indirect measure of cellular organization[11].

Beyond these $T_1$-based approaches, dMRI provides a complementary source of microstructural contrast that is sensitive to complex features, such as fiber orientation, cellular density, and other microscopic tissue properties[34–36]. Increasing evidence suggests that the cortical diffusion signal exhibits depth dependence, revealing microstructural features that are complementary to conventional $T_1$-weighted or quantitative relaxometry methods[53,54]. Even basic diffusion tensor imaging (DTI) shows depth-dependent variations in fractional anisotropy and radial diffusivity within the cortex[53,55,56], while the fiber orientation distribution function (fODF) exhibits robust radial and tangential patterns that vary with cortical curvature and laminar architecture[53,56,57]. These findings underscore the strong potential of dMRI for providing a more comprehensive characterization of cortical cyto- and myeloarchitecture in vivo. Such laminar sensitivity can be further supported by advanced biophysical modeling, which helps disentangle the heterogeneous cellular contributors to the diffusion signal and improves specificity to compartment-level microstructural features reflecting laminar organization[35].

Diffusion MRI enables investigation of tissue microstructural organization in the living human brain by sensitizing the MRI signal to micron-scale Brownian motion of water molecules within tissue[58–60]. The accuracy of imaging tissue micro-geometries using dMRI relies on the maximum gradient strength available during diffusion encoding[61–63]. One of the key technological advances enabling microstructural imaging in the living human brain has been the development of high-performance gradient MRI systems with gradient strengths on the order of hundreds of mT/m for in vivo imaging[27–30]. These greatly exceed the capabilities of conventional clinical MRI scanners, which typically operate with gradient strengths of approximately 40–80 mT/m. The first-generation 3 Tesla human connectome MRI scanner was equipped with a whole-body gradient system capable of reaching a maximum gradient strength ($G_{max}$) of 300 mT/m[64]. This pioneering system enabled major advances in mapping white matter connectivity[65], axonal diameter[66–69], and intra-soma and intra-neurite density across the human lifespan[70], amongst other applications[71–75]. Despite these advances, we and others have shown that pushing gradient strengths beyond 300 mT/m on the original connectome MRI scanner (Connectome 1.0) enables unprecedented in vivo investigations of brain tissue microstructure[61–63].

The recently developed next-generation Connectome MRI scanner (Connectome 2.0) represents a significant leap forward in MRI capabilities for tissue microscopic imaging[27,30]. Equipped with a $G_{max}$ of 500 mT/m and a maximum slew rate ($SR_{max}$) of 600 T/m/s, Connectome 2.0 provides a unique opportunity to probe cortical laminar architecture in the living human brain with greater microstructural sensitivity and spatial resolution than previously attainable. Stronger gradients and higher slew rates bring multiple advantages for microstructural imaging using dMRI[69,76–78], including shorter diffusion times, reduced gradient durations, and shorter echo times (TE), which improves signal-to-noise ratio, spatial resolution, and microstructural specificity to complex tissue properties[27,30,64]. Combined with advanced gray matter biophysical models, such as soma and neurite density imaging (SANDI)[78], diffusion measurements acquired with strong gradients allow disentangling signal contributions from multiple cellular compartments, including the cell body (soma), neurites, and extracellular space. These compartments reflect distinct biological properties of cortical architecture. The intra-neurite fraction is sensitive to axonal and dendritic organization associated with myeloarchitecture, complementing established myelin-sensitive contrasts. The intra-soma fraction provides additional sensitivity to cell body density and cytoarchitectonic organization, offering advantages over existing diffusion MRI contrasts[76,79]. For example, in our recent study of normal brain aging, we applied the SANDI model to data from the Connectome 1.0 scanner and demonstrated age-related reductions in cortical cell body density that closely track cortical volume loss in regions vulnerable to aging[70]. Additionally, in individuals with multiple sclerosis, we observed significant reductions in cortical cell body density within lesions, which correlate with subregional thalamic volume loss[71,72].

The goal of this study was to characterize cortical laminar microstructure in the living human brain by leveraging the enhanced diffusion encoding capabilities of the Connectome 2.0 scanner. We first analyzed SANDI metrics derived from dMRI acquired in healthy young adults, focusing on the supragranular and infragranular layers and performing detailed cortical-depth-dependent analyses to map cytoarchitectonic and myeloarchitectonic organization across cortical depths. Our aim was to uncover laminar and regionally specific variations in microstructure, enabling characterization of cortical areas according to their distinct cytoarchitectonic and myeloarchitectonic properties, and to compare dMRI findings with established histological atlases[8,80,81]. To evaluate the impact of gradient hardware advancements, we also performed a comparative analysis of SANDI metrics obtained using matched protocols on the Connectome 2.0 and Connectome 1.0 scanners, in two cohorts of age- and sex-matched healthy young adults. We hypothesized that the enhanced diffusion

encoding capabilities of the Connectome 2.0 scanner would increase sensitivity to restricted diffusion within cellular compartments, particularly neurites, thereby improving the specificity of in vivo brain microstructure imaging compared with measurements obtained using Connectome 1.0.

## Results

The SANDI model provided parametric maps of six microstructural metrics: intra-soma signal fraction ($f_{is}$), intra-neurite signal fraction ($f_{in}$), extracellular signal fraction ($f_{ec} = 1 − f_{is} − f_{in}$), apparent soma radius ($R_s$), intra-neurite diffusivity ($D_{in}$), and extracellular diffusivity ($D_{ec}$) (Fig. 1).

To assess cortical microstructural features in the living human brain and to relate these depth-dependent patterns to cytoarchitectonic and myeloarchitectonic histological atlases, we applied a framework using a multi-step analysis pipeline (Fig. 2) that includes anatomical parcellation, surface reconstruction, super-resolution image processing of the diffusion-weighted images (2 to 1 mm isotropic; Supplementary Fig. S1) prior to SANDI fitting, deep learning-based supragranular/infragranular segmentation, and laminar sampling of SANDI metrics (intra-soma signal fraction $f_{is}$ and intra-neurite signal fraction $f_{in}$) using FreeSurfer's "mri_vol2surf".

### Cohort characteristics by scanner

A total of 39 healthy adults under 40 years of age participated in this study at Massachusetts General Hospital. Three participants completed scans on both the Connectome 2.0 MRI scanner ($G_{max}$ of 500 mT/m and maximum slew rate of 600 T/m/s) and the Connectome 1.0 MRI scanner ($G_{max}$ of 300 mT/m and maximum slew rate of 200 T/m/s). The Connectome 2.0 cohort consisted of 21 individuals (14 females, 7 males; mean age: 29.0 ± 4.5 years; age range: 19–37) who underwent MRI scans on the newly installed 3T Connectome 2.0 system. The Connectome 1.0 cohort included 21 age- and sex-matched participants (14 females, 7 males; mean age: 28.7 ± 6.2 years; age range: 19–40) who were scanned on the 3T Connectome 1.0 system. The three participants who completed scans on both systems were females aged 27, 32, and 36 years.

### Self-similarity-based super-resolution imaging processing technique

The high-resolution (1 mm isotropic) dMRI-derived SANDI metrics generated using the super-resolution technique[82–84] improved the visualization of detailed microstructural features and effectively reduced partial volume effects compared with the lower-resolution data (Supplementary Figs. S2, S3,

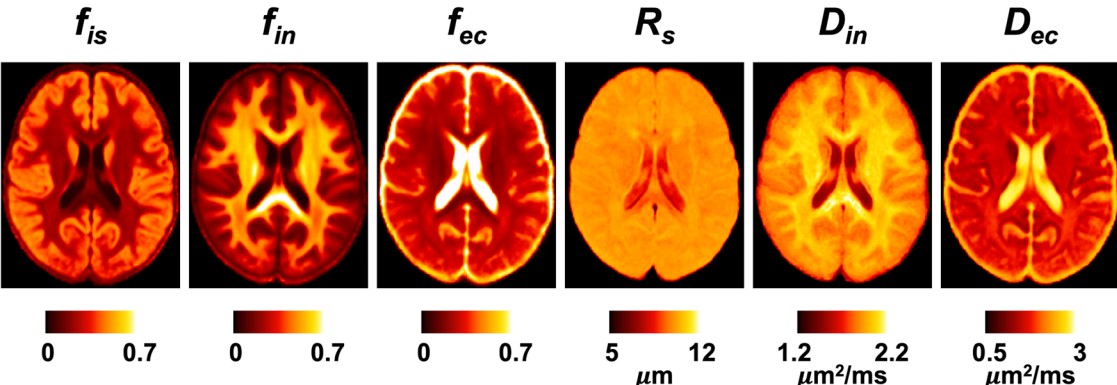

**Fig. 1 | Representative SANDI-derived microstructural metrics, group-averaged across 21 subjects and registered to MNI space.**

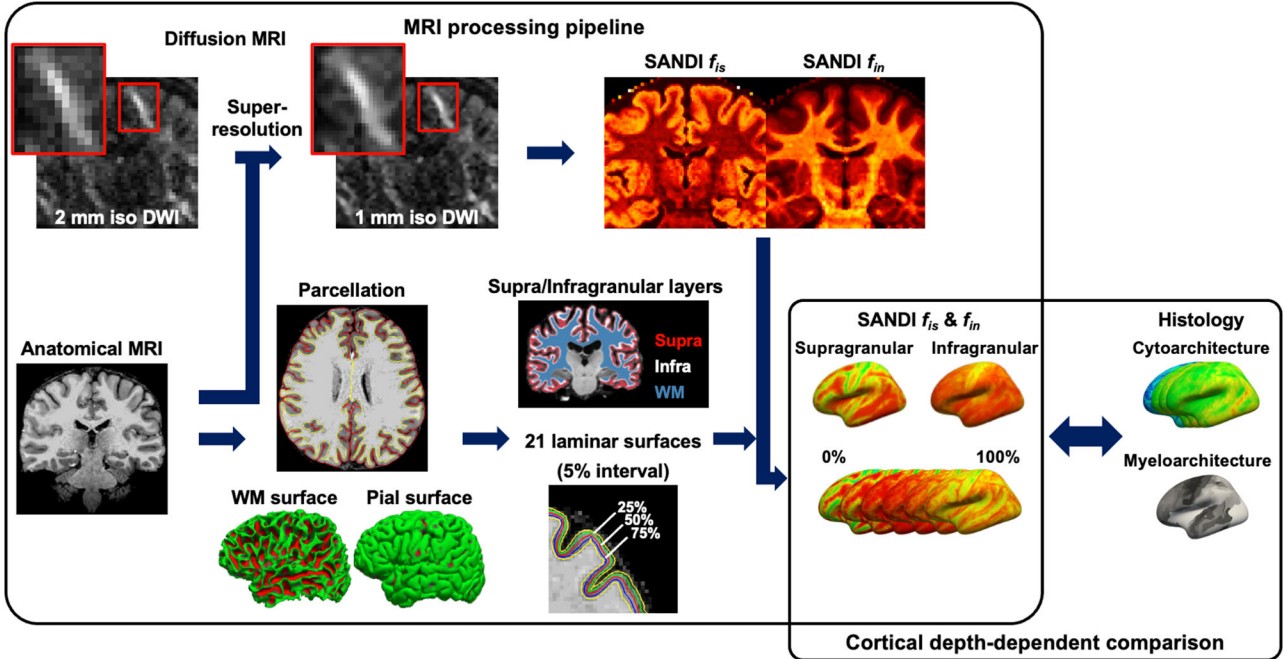

**Fig. 2 | Framework for cortical depth-dependent microstructure analysis of SANDI metrics, with comparisons to histological atlases.**

and see Supplementary Note 1 for details on the super-resolution imaging processing technique).

## SANDI metrics comparison between Connectome 2.0 and Connectome 1.0

The SANDI metrics, averaged across the full cortical depth (0–100%; 5% intervals; pial surface at 0%; white matter surface at 100%) and across subjects for each scanner, are displayed on the FreeSurfer "fsaverage" template space in Supplementary Fig. S5. The intra-neurite signal fraction $f_{in}$ was significantly higher on the Connectome 2.0 scanner (mean = 0.21 ± 0.01) compared to the Connectome 1.0 scanner (mean = 0.16 ± 0.01; false discovery rate (FDR)-$P < 0.001$). In contrast, intra-soma signal fraction $f_{is}$ showed no significant difference between the two scanners (mean = 0.43 ± 0.02 vs. 0.43 ± 0.01; FDR-$P = 0.48$). The three participants who underwent scans on both systems showed the same trends as those observed in the overall sample, with significantly higher intra-neurite signal fraction $f_{in}$ values on Connectome 2.0 (FDR-$P = 0.02$) and no significant differences in intra-soma signal fraction $f_{is}$ (FDR-$P = 0.77$).

## SANDI metrics across cortical depths

Cortical distribution and layer-specific differences in microstructural metrics derived from the SANDI model on the Connectome 2.0 scanner are illustrated in Fig. 3. Cortical surface maps showed the intra-soma signal fraction $f_{is}$ and intra-neurite signal fraction $f_{in}$ across the entire cortex (overall), as well as separately within supragranular (supra) and infragranular (infra) layers. These laminar profiles revealed significantly higher intra-soma signal fraction $f_{is}$ and intra-neurite signal fraction $f_{in}$ in the infragranular layers (mean $f_{is}$ = 0.44 ± 0.02, mean $f_{in}$ = 0.24 ± 0.01) compared to the supragranular layers (mean $f_{is}$ = 0.41 ± 0.02, mean $f_{in}$ = 0.17 ± 0.01), with both comparisons showing FDR-$P < 0.001$.

The utility of laminar-specific metrics to distinguish adjacent cortical regions is illustrated in Supplementary Fig. S6. While the overall intra-soma signal fraction $f_{is}$ did not differ significantly between motor cortex sub-regions BA4a and BA4p (BA4a: 0.39 ± 0.03, BA4p: 0.39 ± 0.03; FDR-$P$ = n.s.), layer-specific comparisons uncovered significant differences between supra- and infragranular estimates of intra-soma signal fraction $f_{is}$. In the supragranular layer, intra-soma signal fraction $f_{is}$ was significantly lower in BA4a compared to BA4p (mean = 0.36 ± 0.03 vs. 0.39 ± 0.04; FDR-$P < 0.001$). Conversely, infragranular intra-soma signal fraction $f_{is}$ was higher in BA4a than in BA4p (mean = 0.41 ± 0.02 vs. 0.40 ± 0.03; FDR-$P < 0.001$).

Each SANDI metric projected onto the cortical surface template displayed distinct microstructural patterns across 21 cortical depths, offering detailed insights into layer-specific microstructural organization. Figure 4 presents the intra-soma signal fraction $f_{is}$ from the Connectome 2.0 scanner at different depths alongside the Merker staining intensity reflecting cell body density obtained from the BigBrain cytoarchitectonic atlas. Both profiles show the lowest values near the pial surface, followed by a gradual increase with increasing depth. Across all cortical regions, the intra-soma signal fraction $f_{is}$ reached its peak at approximately 55% cortical depth, while the Merker staining intensity from the BigBrain cytoarchitectonic atlas peaked at around 63% depth. Additionally, both equidistant and equivo-lumetric sampling schemes in the LayNii laminar-fMRI toolbox[85] yielded a peak in the intra-soma signal fraction $f_{is}$, at the same depth (~55%) (Supplementary Fig. S7).

The intra-neurite signal fraction $f_{in}$ from the Connectome 2.0 scanner showed a progressive increase from the pial surface to the gray-white matter boundary, consistent with established histological observations (Fig. 5). Among cortical areas, the sensorimotor and auditory cortices exhibited the highest intra-neurite signal fraction $f_{in}$, in agreement with the myeloarchitectonic atlas (highlighted by red arrows). Furthermore, the intra-neurite signal fraction $f_{in}$ in the infragranular layer was negatively correlated with myelin staining intensity across regions defined by Nieuwenhuys' parcellation ($r = -0.22$; $P = 0.002$). The Connectome 1.0 data preserved the depth-dependent $f_{in}$ profile, showing an increase toward the white matter with overall lower values and a correlation with myeloarchitecture patterns ($r = -0.18$; $P = 0.002$) comparable to that observed on Connectome 2.0 (Supplementary Fig. S8).

## Regional differences in intra-soma signal fraction $f_{is}$ across cortical depths

When comparing the motor and visual cortices across multiple cortical depths (Fig. 6), the intra-soma signal fraction $f_{is}$ from the Connectome 2.0 scanner showed distinct depth-dependent profiles between the two regions. The visual cortex exhibited higher intra-soma signal fraction $f_{is}$ than the motor cortex at several depths, particularly in the deeper portions of the cortical depths. Specifically, the intra-soma signal fraction $f_{is}$ was significantly higher in the visual cortex at 0% (FDR-$P = 0.002$) and across the 65–100% depths (FDR-$P < 0.009$), a pattern aligned with both cytoarchitectonic layer profiles[86] and Merker staining intensity from the BigBrain atlas. In addition to differences in magnitude, the two regions differed in the depth at which the intra-soma signal fraction $f_{is}$ peaked. The motor cortex showed a shallower peak at ~40% depth (Fig. 6c), whereas the visual cortex exhibited a deeper peak at ~65%. This depth shift is reflected in the BigBrain cell body staining intensity profiles, which show a peak at ~53% depth in the motor cortex and ~67% in the visual cortex.

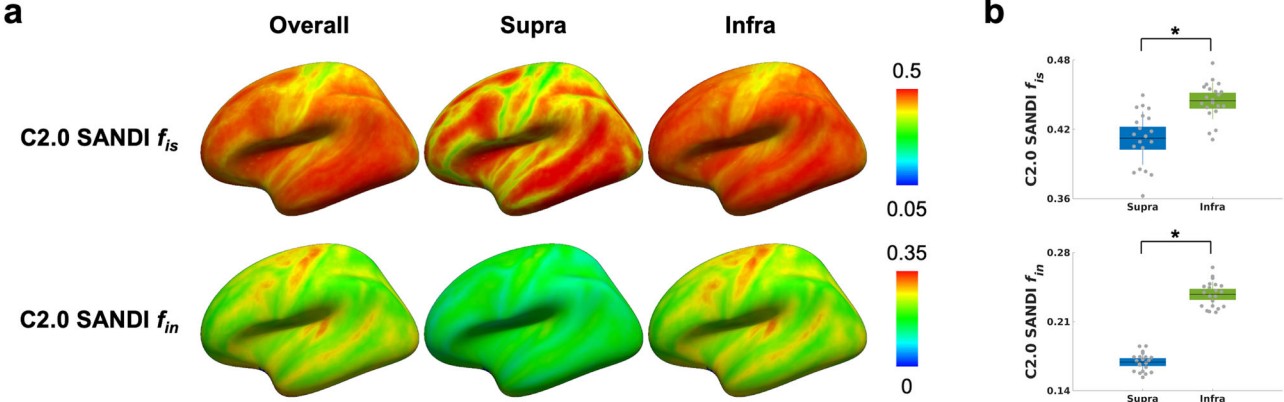

**Fig. 3 | SANDI-derived microstructural metrics on the Connectome 2.0 scanner in supragranular (supra) and infragranular (infra) layers. a** Cortical surface maps of the intra-soma signal fraction $f_{is}$ and intra-neurite signal fraction $f_{in}$ derived from SANDI, averaged across 21 individuals. **b** Boxplots summarizing intra-soma signal fraction $f_{is}$ and intra-neurite signal fraction $f_{in}$ across 21 individuals, with statistically significant differences indicated (∗: FDR-$P < 0.05$) between supragranular and infragranular layers. The box represents the 95% confidence interval (mean ± 1.96 standard error of the mean). The solid horizontal line indicates the mean, and the thinner vertical lines denote ± 1 standard deviation. Individual dots correspond to individual subjects. C2.0 Connectome 2.0.

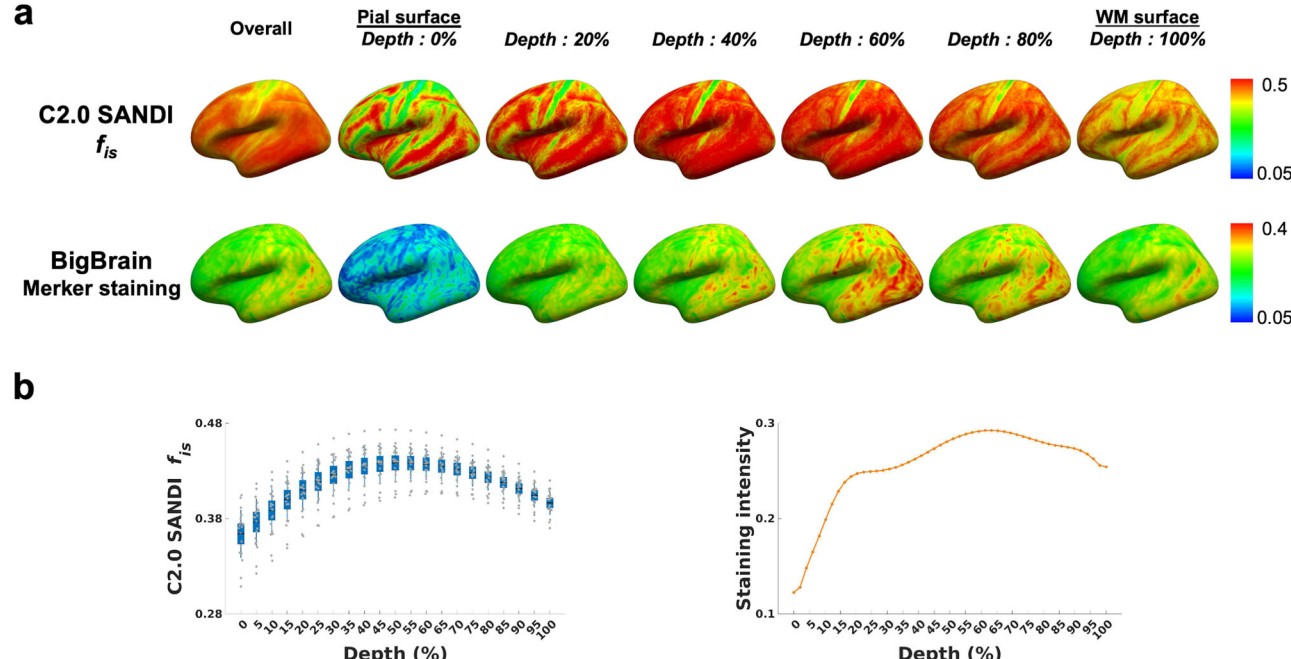

**Fig. 4 | Cortical depth-dependent intra-soma signal fraction $f_{is}$ from the Connectome 2.0 scanner compared with Merker staining intensity from the BigBrain atlas. a** Cortical surface maps of the intra-soma signal fraction $f_{is}$ from the Connectome 2.0 scanner and Merker staining intensity from the BigBrain atlas, shown for the overall cortex and across cortical depths from the pial surface (0%) to the white matter surface (100%). **b** Cortical depth profiles of intra-soma signal

fraction $f_{is}$ and Merker staining intensity. For the intra-soma signal fraction, $f_{is}$, boxplots summarize data from 21 individuals. The box represents the 95% confidence interval (mean ± 1.96 standard error of the mean). The solid horizontal line indicates the mean, and the thinner vertical lines denote ± 1 standard deviation. Individual dots correspond to individual subjects. Data on Merker staining intensity are sourced from Paquola et al.[7]. C2.0 Connectome 2.0.

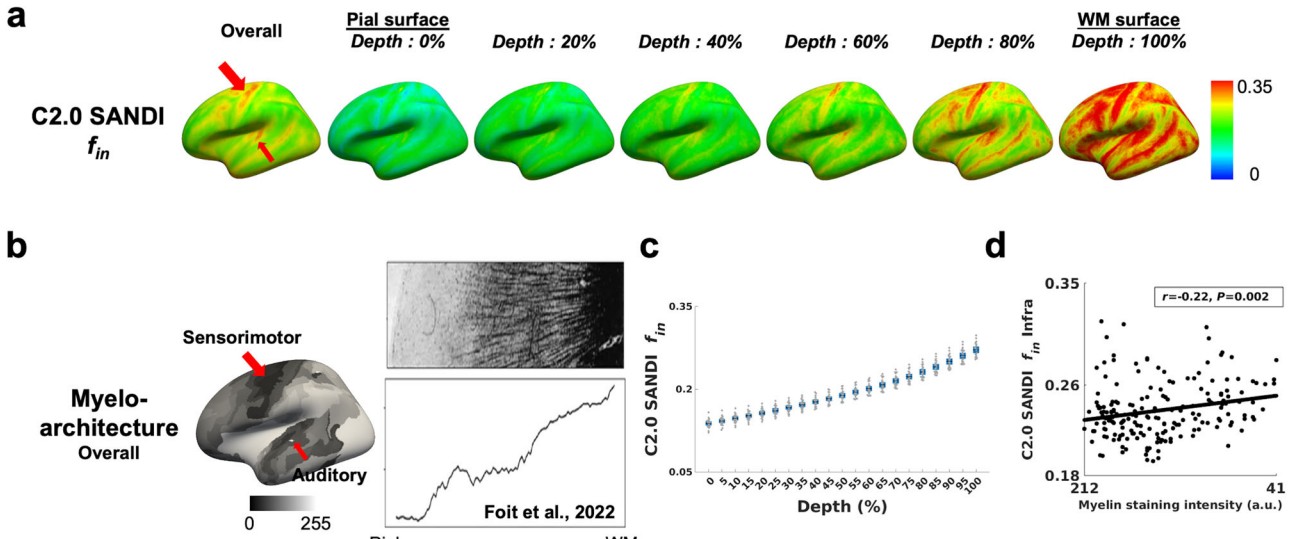

**Fig. 5 | Cortical depth-dependent intra-neurite signal fraction $f_{in}$ from the Connectome 2.0 scanner in comparison with myelin staining intensity from the myeloarchitecture atlas. a** Cortical surface maps of the intra-neurite signal fraction $f_{in}$ from the Connectome 2.0 scanner, shown for the overall cortex and across cortical depths from the pial surface (0%) to the white matter surface (100%).
**b** Myeloarchitecture map across the overall cortex, together with a representative histological image of myelin staining and the corresponding cortical depth profile from the myeloarchitecture atlas. Darker colors (i.e., lower intensity values) correspond to higher myelin concentration across regions defined by Nieuwenhuys'

parcellation. **c** Cortical depth profile of intra-neurite signal fraction $f_{in}$ across 21 individuals. The box represents the 95% confidence interval (mean ± 1.96 standard error of the mean). The solid horizontal line indicates the mean, and the thinner vertical lines denote ± 1 standard deviation. Individual dots correspond to individual subjects. **d** Association between regional intra-neurite signal fraction $f_{in}$ and myelin staining intensity across cortical regions. The solid black line represents the linear fit. Myelin staining intensity data and histological images are sourced from Foit et al.[8], with permission from Elsevier. C2.0 Connectome 2.0.

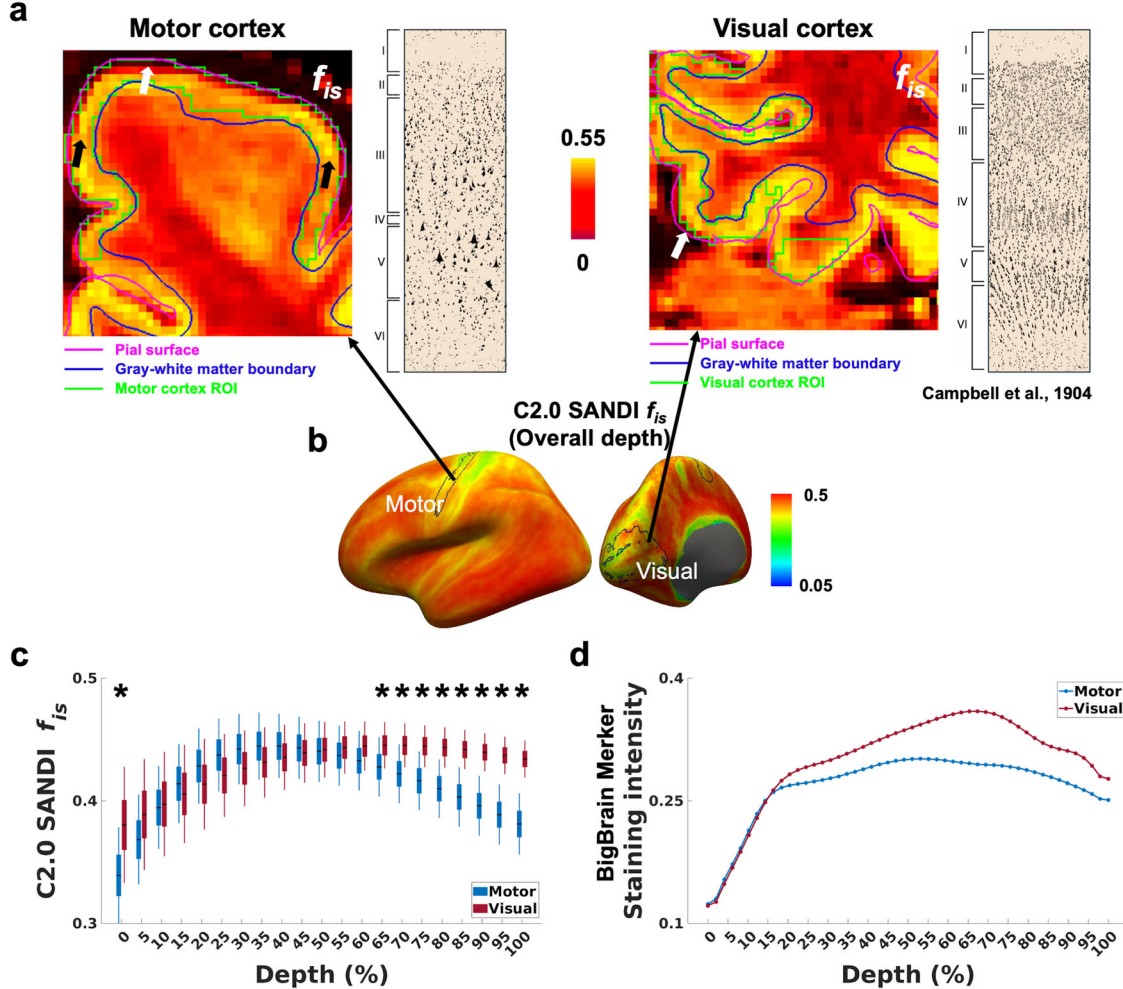

**Fig. 6 | Cortical depth-dependent differences in intra-soma signal fraction $f_{is}$ between the motor and visual cortices. a** Representative intra-soma signal fraction $f_{is}$ maps from motor and visual cortices, derived from the Connectome 2.0 scanner. Corresponding cytoarchitectonic layer profiles are shown from Campbell AW (1904)[86]. **b** Whole-cortex surface maps indicating the motor and visual regions across overall depths. **c** Cortical depth profiles of intra-soma signal fraction $f_{is}$ for motor and visual cortices, summarized across 21 individuals. The box represents the

95% confidence interval (mean ± 1.96 standard error of the mean). The solid horizontal line indicates the mean, and the thinner vertical lines denote ± 1 standard deviation. Individual dots correspond to individual subjects. Statistically significant differences are indicated (∗: FDR-$P < 0.05$). **d** Cortical depth profiles of Merker staining intensity from the BigBrain atlas for motor and visual cortices. Data are sourced from Paquola et al.[7]. C2.0 Connectome 2.0.

The intra-soma signal fraction $f_{is}$ profiles across von Economo's cytoarchitectonic cortical types revealed depth-dependent patterns unique to each cortical class (agranular, frontal, parietal, polar, and granular) (Supplementary Fig. S9). Agranular and frontal cortices exhibited peak values at relatively shallow cortical depth (~45%), whereas granular cortex presented a deeper peak (~65%), reflecting the prominent layer IV in the granular cortex, as described in the laminar cytoarchitectonic features of von Economo's histological atlas[87].

### Relationship between intra-soma signal fraction $f_{is}$ and cortical curvature

The intra-soma signal fraction $f_{is}$ exhibited a layer-dependent curvature-microstructure relationship, as shown in Fig. 7. In the supragranular layer, a positive relationship was observed ($r = 0.58$; $P < 0.001$), indicating higher intra-soma signal fraction $f_{is}$ in sulcal fundi compared to gyral crowns (Supplementary Fig. S10 for representative maps). In contrast, this relationship reversed in the deeper cortex: the infragranular layer showed a weak negative correlation ($r = -0.05$; $P = 0.74$), and a stronger negative correlation emerged at 90% cortical depth ($r = -0.38$; $P = 0.003$), indicating higher intra-soma signal fraction $f_{is}$ in the gyral crowns than in the sulcal

fundi at deeper cortical layers, consistent with previous histological observations[88] (Supplementary Fig. S11). These relationships were preserved when considering only vertices with cortical thickness ≥ 2 mm (Supplementary Fig. S12).

### Discussion

Our work shows that fitting the SANDI model to diffusion MRI data acquired with the 500 mT/m gradient system of the next-generation Connectome 2.0 scanner enables in vivo characterization of cortical microstructure in the human brain, allowing resolution of cytoarchitectonic and myeloarchitectonic features across cortical depths. We identified distinct laminar and regional microstructural profiles that closely correspond to histological patterns reported in established atlases[8,80,81,87] and demonstrate improved sensitivity compared with measurements obtained on the Connectome 1.0 scanner. Specifically, we observed layer-specific patterns in SANDI-derived microstructural metrics, including a peak in intra-soma signal fraction $f_{is}$ at mid-cortical depth and increasing intra-neurite signal fraction $f_{in}$ toward the gray-white matter boundary. We also identified regionally specific variations, such as higher intra-soma signal fraction $f_{is}$ in the visual cortex compared with the motor cortex, particularly at deeper

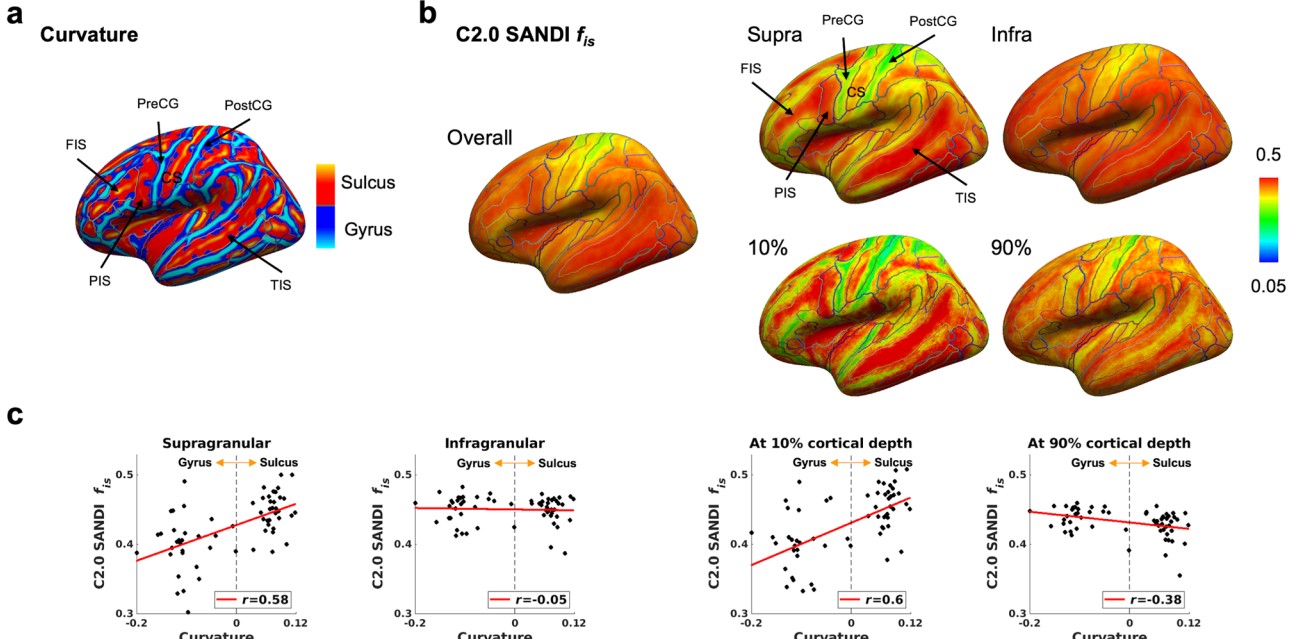

**Fig. 7 | Association between intra-soma signal fraction $f_{is}$ and cortical curvature across cortical depths. a** Cortical surface map of curvature with anatomical labels from the aparc.a2009s.annot parcellation. **b** Cortical surface maps of the intra-soma signal fraction $f_{is}$ from the Connectome 2.0 scanner, shown for the overall cortex, supragranular (supra) and infragranular (infra) layers, and at 10 and 90% cortical depths. **c** Associations between cortical curvature and intra-soma signal fraction $f_{is}$ across regions for supragranular and infragranular layers, and at 10 and 90% cortical depths. Each dot represents a cortical region. Solid red lines indicate linear fits. Reported values denote Pearson correlation coefficients ($r$). C2.0 Connectome 2.0. CS central sulcus, FIS frontal inferior sulcus, PIS parietal inferior sulcus, PostCG postcentral gyrus, PreCG precentral gyrus, TIS temporal inferior sulcus.

cortical depths. In addition, a depth-dependent relationship between intra-soma signal fraction $f_{is}$ and cortical curvature emerged, with higher $f_{is}$ values in sulcal regions within the supragranular layers that shift toward gyral regions in deeper cortical layers. Collectively, these findings demonstrate the promise of high-performance gradient diffusion MRI for detailed in vivo mapping of cortical microstructure, helping to bridge the gap between traditional histology and noninvasive neuroimaging, with broad potential applications in neuroscience, neurology, and psychiatry. For example, the approaches presented here may enable detection of subtle alterations in cellular architecture across cortical layers and columns associated with aging and neurodegenerative disease. More fundamentally, if architectonic organization of the cortex predicts patterns of connectivity[87,89–92], then the ability to map tissue microstructure across cortical layers may enable the development of microstructure profile-based connectomes, thereby advancing an understanding of how the human brain is organized and wired across spatial scales[89].

### Enhanced microstructural sensitivity with Connectome 2.0

The Connectome 2.0 scanner demonstrates enhanced microstructural sensitivity compared with Connectome 1.0, enabling more precise characterization of both neurite and soma compartments. The stronger gradient system permits substantially shorter diffusion and echo times, which reduce the impact of inter-compartmental water exchange effects[74,93–95], that are not explicitly modeled in SANDI[78], and improve SNR, resulting in higher estimated intra-neurite signal fraction $f_{in}$ values. Beyond neurites, although whole-cortex averages of the intra-soma signal fraction $f_{is}$ did not differ substantially between scanners, more detailed subregional analyses using Connectome 2.0 revealed improved performance of soma-related SANDI metrics. Specifically, the intra-soma signal fraction $f_{is}$ showed greater regional contrast across Brodmann areas (BA4, 3a, 3b, and 1), closely aligning with prior histological observations, and exhibited reduced inter-subject variability compared with Connectome 1.0[30]. Furthermore, Connectome 2.0 demonstrated greater sensitivity to smaller soma populations,

highlighting its potential for more refined characterization of cortical cytoarchitecture.

### Laminar and regional variations in cortical microstructure with histological correspondence

Cortical depth-dependent dMRI analysis offers a more detailed characterization of cortical regions[53–57], unveiling microstructural information that may be overlooked by conventional ROI-based methods. SANDI metrics obtained with high-performance gradient systems revealed distinct profiles across cortical depths, enabling noninvasive characterization of cortical laminar organization. The intra-soma signal fraction $f_{is}$ and intra-neurite signal fraction $f_{in}$ exhibited patterns that qualitatively correspond to known histological features of cortical layers. Specifically, our results showed a peak in intra-soma signal fraction $f_{is}$ at mid-cortical depth, corresponding closely to the peak in cell body density reported in the BigBrain atlas[7], which occurred at 63% depth in layers enriched with large pyramidal neurons and higher cell body density. Intra-neurite signal fraction $f_{in}$ progressively increased toward deeper cortical layers, consistent with histological data showing higher myelinated fiber density near the gray-white matter boundary[8].

Region-specific variations in SANDI metrics further highlight the value of depth-dependent analysis, as cytoarchitectonic and myeloarchitectonic properties vary substantially across cortical regions. The primary motor cortex comprises two cytoarchitectonic subdivisions, BA4a and BA4p[14,96], and depth-dependent intra-soma signal fraction $f_{is}$ successfully differentiated these adjacent regions, revealing distinct supragranular and infragranular profiles that were not apparent with conventional whole-thickness measures. Specifically, BA4a showed lower intra-soma signal fraction $f_{is}$ in the supragranular layers but higher intra-soma signal fraction $f_{is}$ in the infragranular layers relative to BA4p.

Broader regional distinctions were also evident. The primary motor cortex lacks a well-defined granular layer (layer IV), deviating from the classical six-layered structure observed in somatosensory cortices[9], while the

primary visual cortex is marked by the line of Gennari, a dense band of myelination within layer IV[3,6]. Reflecting these known distinctions, the visual cortex exhibited higher intra-soma signal fraction $f_{is}$ than the motor cortex, particularly at deeper cortical depths (65–100%), consistent with cell density patterns reported in histological studies[97–99] and with Merker staining intensity in the BigBrain atlas.

To provide broader histological context for these region- and depth-dependent patterns, we examined intra-soma signal fraction $f_{is}$ profiles across von Economo's cytoarchitectonic cortical types[87]. Distinct laminar profiles emerged across these types. Agranular cortices (including the primary motor cortex) and frontal cortices showed relatively shallow peaks (~45% depth), reflecting their reduced or absent layer IV and the predominance of larger pyramidal neurons in layers III and V. In contrast, granular cortices (including primary visual, somatosensory, and auditory cortices) exhibited deeper peaks (~65% cortical depth), consistent with their well-developed layer IV, which contains a dense population of small granule cells that markedly contribute to the soma signal fraction $f_{is}$. These patterns closely align with the classical laminar architecture described in von Economo's histological atlas.

SANDI-derived intra-neurite signal fraction $f_{in}$ further extended these depth- and region-dependent patterns. The highest intra-neurite signal fraction $f_{in}$ values were detected in primary sensory cortices, including somatosensory and auditory regions, as well as in the primary motor cortex. These findings align with neurite density estimates derived from another biophysical dMRI model, neurite orientation dispersion and density imaging (NODDI), and with patterns reported in the myeloarchitectonic atlas[8,100]. These observations were further supported by a significant negative correlation between intra-neurite signal fraction $f_{in}$ and myelin staining intensities across regions defined by Nieuwenhuys' cortical parcellation. However, the magnitude of this correlation was lower than expected, likely due to the definition of intra-neurite signal fraction $f_{in}$ in the SANDI model, which captures water diffusion within both myelinated and unmyelinated axons as well as dendrites. The inclusion of unmyelinated intra-neurite components reduces the specificity of the intra-neurite signal fraction $f_{in}$ to myelinated fiber content, thereby attenuating the observed correlation with histological myelin density.

## Cortical geometry (curvature) and laminar organization

Our findings revealed a layer-specific relationship between intra-soma signal fraction $f_{is}$ and cortical curvature, providing additional evidence for laminar variation in cytoarchitecture. Specifically, intra-soma signal fraction $f_{is}$ was positively correlated with curvature in the supragranular layer, indicating higher intra-soma signal fraction $f_{is}$ values in sulcal fundi compared to gyral crowns. In contrast, intra-soma signal fraction $f_{is}$ showed a negative correlation with curvature in deeper cortical regions, where higher intra-soma signal fraction $f_{is}$ values were observed in gyral regions. These results were further confirmed by a two-sample t-test between the sulcus and gyrus, with corresponding $P < 0.001$ at 10% depth and $P = 0.008$ at 90% depth. This inversion pattern is consistent with previous histological studies reporting neuronal density in supragranular and infragranular layers of gyri and sulci[88]. While our label-based analysis used the aparc.a2009s parcellation to define sulcal and gyral regions, curvature alone does not always align perfectly with anatomical definitions. For example, secondary folds at the base of deep sulci may exhibit locally positive curvature, typically associated with gyral crowns, despite being embedded within clearly sulcal cortex. Whether such regions more closely resemble sulcal or gyral cortex in their microstructural and cytoarchitectonic features remains an open question. This anatomical ambiguity should be taken into account when interpreting vertex-based analyses of curvature and its relationship to laminar architecture.

## Clinical translation and future scientific directions

Advances in gradient hardware and biophysical modeling of the dMRI signal are poised to accelerate translation into both clinical applications and basic neuroscience research. Although the ability to characterize layer-specific cytoarchitecture and myeloarchitecture noninvasively holds great promise, widespread implementation remains limited by the high gradient strengths required, which are not yet standard on most clinical MRI systems. Nonetheless, recent studies have demonstrated the feasibility of applying the SANDI model on conventional 3T scanners, such as the Philips Ingenia CX and Siemens Prisma scanners, indicating that meaningful microstructural information can still be extracted in patient populations despite hardware constraints[101,102]. Importantly, our results further support the potential translatability of this framework. Although the Connectome 2.0 scanner provides greater microstructural sensitivity than its predecessor[30], the Connectome 1.0 scanner still retains sufficient sensitivity to capture depth-dependent cortical microstructural organization. These findings support the feasibility of implementing the laminar SANDI framework on emerging high-performance clinical MRI systems, such as the Siemens 3T Cima.X and GE MAGNUS scanners[28], and potentially more broadly on currently available and future 3T clinical MRI platforms.

Continued efforts are needed to optimize acquisition strategies, improve model robustness, and validate derived microstructural metrics across diverse populations and disease conditions. Since cortical fODFs exhibit distinct radial and tangential organization across cortical laminae[53], integrating fODF-based orientation information with SANDI-derived compartment metrics may enhance the laminar sensitivity of dMRI-based cortical characterization. In parallel, advances in MRI-based cortical layer segmentation are expected to strengthen further the translational potential of the laminar SANDI framework[11,37]. Incorporating layer-specific segmentation derived from quantitative T1 mapping could enable more precise assignment of SANDI-derived microstructural metrics to specific cortical layers[46] and provide a more direct in vivo surrogate for cytoarchitectonic and myeloarchitectonic organization. Further improvements in super-resolution reconstruction and high-resolution submillimeter imaging techniques will also be essential for enhancing laminar sampling sensitivity and minimizing partial volume effects. Ultimately, these advances may facilitate the clinical adoption of microstructural dMRI approaches for early diagnosis and longitudinal monitoring of neurodegenerative and psychiatric disorders.

## Limitations

To bridge the gap between in vivo imaging and histological ground truth, we compared SANDI-derived microstructural metrics with established cytoarchitectonic and myeloarchitectonic atlases. Although our results demonstrated strong agreement between SANDI metrics and histological references, suggesting that SANDI can approximate the cortical microstructural features reflected in histology, it is important to note the limitations in establishing a direct, one-to-one correspondence between the two modalities. The histological references used in this study, the BigBrain atlas for cytoarchitecture and the Nieuwenhuys atlas for myeloarchitecture, capture laminar patterns but do not fully represent population-level variability. In particular, the BigBrain atlas is derived from a single 65-year-old brain, limiting generalizability[80]. Additionally, discrepancies between the in vivo MRI surfaces and the histological surfaces can introduce depth-dependent misalignment. In the BigBrain atlas, the gray-white matter boundary is delineated from cell body staining contrast, whereas FreeSurfer MRI surfaces rely on $T_1$-weighted intensity, which is primarily driven by myelin. These differences in boundary definitions likely contributes to the observed 55% versus 63% peak depth discrepancy between SANDI-derived intra-soma signal fraction $f_{is}$ and BigBrain Merker cell density profiles. Sampling near the pial surface and the gray-white matter boundary is further complicated by partial volume effects from CSF, local curvature differences, and the resolution gap between ultra-high-resolution (20 μm) BigBrain data and 1 mm in vivo dMRI.

While the SANDI model provides valuable insights into cortical microstructure, it has inherent limitations in modeling microstructural properties. The model's signal fractions are relative estimates derived from

complex mathematical representations of water diffusion across three compartments, influenced by $T_2$ weighting and linear interdependence among compartments, i.e., the sum of all signal fractions is 1[103]. A key challenge involves distinguishing between intra-soma and extracellular water compartments that share similar geometric models but exhibit different diffusivities. Additionally, the current model cannot fully account for the presence of free water or partial volume effects from adjacent cerebrospinal fluid, which can lead to biased estimates, particularly near cortical boundaries or pathologies with edema and inflammation[77,104]. Furthermore, partial volume effects arising from cortical folding, especially in highly curved regions, and limited spatial resolution can lead to mixing of signals across adjacent layers or tissue compartments, potentially biasing compartment-specific signal fraction estimates at the vertex level. To improve specificity, future research should focus on refining the model to better separate extracellular compartments (intermediate diffusivity, hindered diffusion) from soma (low diffusivity, restricted diffusion) and free water (high diffusivity, free diffusion), thereby reducing partial volume effects and enabling more accurate capture of the cellular characteristics in the normal cortex and in pathological conditions.

## Conclusions

Our study demonstrates the potential of in vivo cortical gray matter microstructural imaging by combining ultra-high-performance gradient MRI with the SANDI diffusion MRI biophysical model. Using the next-generation Connectome 2.0 scanner, cortical depth-dependent analysis of SANDI metrics enables detailed characterization of cortical cytoarchitecture and myeloarchitecture in the living human brain, revealing distinct laminar profiles that closely align with established histological patterns, particularly in the distribution of cell body density and myelinated fibers across cortical layers.

Overall, our findings provide further evidence that high-performance gradient MRI systems can help bridge the gap between traditional postmortem histology and in vivo neuroimaging, opening new opportunities for developing noninvasive biomarkers capable of detecting subtle, layer-specific microstructural changes associated with a broad range of neurological and psychiatric disorders.

## Methods and materials
### Participant recruitment

We recruited healthy young adults between the ages of 19 and 40 years for scans on the Connectome 1.0 and Connectome 2.0 scanners. Participants for the Connectome 1.0 scans were recruited between September 2016 and June 2023. After the Connectome 1.0 scanner was decommissioned and replaced by the Connectome 2.0 scanner in the same imaging scanner bay, a new group of participants was recruited for scans on the Connectome 2.0 scanner between September 2023 and October 2024. A subset of participants ($N = 3$) were recruited for both scanners within inter-scan intervals of 4–11 months. Our screening process excluded individuals with any history of neurological and psychiatric conditions, encompassing conditions such as dementia, cerebrovascular disease, brain tumors, head injuries, and any other central nervous system disorders. All subjects provided written informed consent prior to participation. The research protocols were reviewed and approved by the Institutional Review Board of Mass General Brigham and were conducted in accordance with the Declaration of Helsinki. All ethical regulations relevant to human research participants were followed.

### Data acquisition

All MRI scans were performed at the Athinoula A. Martinos Center for Biomedical Imaging, Massachusetts General Hospital. The 3T Connectome 2.0 MRI scanner (MAGNETOM Connectom. X, Siemens Healthineers, Erlangen, Germany) is equipped with a $G_{max}$ of 500 mT/m and an $SR_{max}$ of 600 T/m/s, using a custom-built 72-channel in vivo head coil for signal reception[105]. The 3T Connectome 1.0 MRI scanner (MAGNETOM Connectom, Siemens Healthcare) was equipped with a $G_{max}$ of 300 mT/m and

$SR_{max}$ of 200 T/m/s using a custom-built 64-channel in vivo head coil[106]. On both scanners, dMRI data were acquired using a pulsed gradient spin-echo echo-planar-imaging (EPI) sequence. The diffusion times were set to the minimum accessible values for each system for the maximum $b$-value of 6000 s/mm$^2$, with $\Delta = 13$ ms on the Connectome 2.0 scanner and 19 ms on the Connectome 1.0 scanner, respectively, and diffusion-weighted gradient durations ($\delta$) of 6 ms and 8 ms, respectively. A total of eight $b$-values were linearly sampled in gradient strength up to $G_{max}$, with 32 diffusion encoding directions for $b < 2400$ s/mm$^2$ ($b = 50, 350, 800$, and 1500 s/mm$^2$) and 64 directions for $b \geq 2400$ s/mm$^2$ ($b = 2400, 3450, 4750$, and 6000 s/mm$^2$) uniformly distributed on a sphere. Interspersed non-diffusion-weighted images ($b = 0$ s/mm$^2$) were obtained for every 16 diffusion-weighted images to normalize signal intensity. The repetition time/echo time (TR/TE) were 3600/53 ms for the Connectome 2.0 scanner and 4000/77 ms for the Connectome 1.0 scanner. The imaging planes for the Connectome 2.0 scanner were axial, whereas those for the Connectome 1.0 scanner were sagittal. Additional common parameters for both scanners included: 2 mm isotropic voxel size, partial Fourier = 6/8, generalized autocalibrating partially parallel acquisition (GRAPPA) acceleration factor = 2, simultaneous multislice (SMS) acceleration factor = 2, anterior-to-posterior phase encoding direction, and adaptive coil combination. To correct for susceptibility-induced distortion, we acquired ten additional non-diffusion-weighted images at the beginning of the dMRI scans with a reversed-phase encoding direction (posterior-to-anterior).

For cortical surface reconstruction and segmentation, high-resolution 3D $T_1$-weighted anatomical images were acquired during the same session. For the Connectome 2.0 scanner, a magnetization-prepared rapid acquisition with gradient echo (MPRAGE) sequence was employed with the imaging parameters: 1 mm isotropic voxel size, TR/TE = 2500/3.36 ms, TI = 1100 ms, flip angle = 8°, and GRAPPA acceleration factor = 2. For the Connectome 1.0 scanner, we used a multi-echo magnetization-prepared rapid acquisition with gradient echo (MEMPRAGE) sequence with the following parameters: 1 mm isotropic voxel size, TR/TE = 2530/1.15, 3.03, 4.89, and 6.75 ms, TI = 1100 ms, flip angle = 7°, and GRAPPA acceleration factor = 3.

### Data processing

dMRI data were preprocessed using an in-house script based on the DESIGNER pipeline[107]. Raw dMRI data were corrected for Gibbs ringing artifact using the "mrdegibbs" function in MRtrix3[108,109], susceptibility and eddy current-induced distortions using the "topup" and "eddy" functions in FSL[110,111], followed by a gradient nonlinearity correction[75]. A self-similarity-based super-resolution image processing technique was applied to dMRI data by introducing high-resolution details from $T_1$-weighted anatomical images[82–84], resulting in 1 mm high-resolution dMRI data to reduce partial volume effects (see Supplementary Note 1 for details). The SANDI model was fitted to multi-shell dMRI signals averaged over gradient directions (spherical mean), employing the SANDI MATLAB toolbox (https://github.com/palombom/SANDI-Matlab-Toolbox-v1.0)[78], with intrinsic soma diffusivity ($D_{is}$) fixed at 2 μm$^2$/ms[112]. To assess the robustness of parameter estimation, a noise propagation analysis was conducted using simulated signals incorporating Rician noise as shown in the Supplementary Note 2 and Supplementary Fig. S4. Resulting SANDI parametric maps comprise the intra-soma signal fraction ($f_{is}$), intra-neurite signal fraction ($f_{in}$), extracellular signal fraction ($f_{ec}$), apparent soma radius ($R_s$), intra-neurite diffusivity ($D_{in}$), and extracellular diffusivity ($D_{ec}$). In this study, the intra-soma signal fraction $f_{is}$ and intra-neurite signal fraction $f_{in}$ were of particular interest, as they reflect distinct aspects of the underlying tissue microstructure. Specifically, intra-soma signal fraction $f_{is}$ reflects cytoarchitectonic features, such as cell body density and organization, while intra-neurite signal fraction $f_{in}$ is associated with the myeloarchitecture, particularly the distribution of aligned neurites, including axons and dendrites.

We processed 3D $T_1$-weighted anatomical images using FreeSurfer (version 7.1.4, https://surfer.nmr.mgh.harvard.edu) through the standard "recon-all" pipeline for skull stripping, cortical gray matter parcellation, and cortical surface reconstruction[113]. From this pipeline, we obtained cortical curvature maps and the aparc.a2009s.annot parcellation, as well as additional surface-based labels for regions of interest. The primary motor cortex was defined based on Brodmann's area 4, comprising its anterior (4a) and posterior (4p) subdivisions. The visual cortex was delineated using Brodmann area 17 (primary visual cortex, V1) and area 18 (secondary visual cortex, V2). To further characterize the laminar profiles with region-specific cytoarchitectonic information, cortical regions were additionally classified according to von Economo's cortical types using the ENIGMA toolbox[114], which provides surface-based parcellations of the agranular, frontal, parietal, polar, and granular classes on the FreeSurfer "fsaverage" cortical surface template.

The averaged non-diffusion-weighted image was aligned with the $T_1$-weighted anatomical image using the "bbregister" function in FreeSurfer, which employs a boundary-based rigid body transformation with 6 degrees of freedom. To investigate laminar patterns of cortical organization, the cortex was segmented into supragranular layer and infragranular layer using a deep learning-based approach guided by cortical curvature and gray matter structure[115], with surface placement based on an optimized iso-volume model (https://github.com/simnibs/cortech). A cascaded multi-resolution U-Net was trained on in vivo and ex vivo MRI channels, with weak supervision applied to the in vivo data to ensure that the combined supragranular and infragranular layers fully covered the cortical gray matter labels. We then utilized the FreeSurfer commands "mri_compute_layer_fractions" and "mri_compute_layer_intensities" to extract SANDI-derived metrics across the whole cortical thickness as well as within supragranular and infragranular layers. In addition, the cortex was further divided into 21 evenly spaced depth intervals (5% intervals), ranging from the pial surface (0% depth) to the white matter boundary (100% depth), for a depth-dependent analysis of microstructural features. SANDI metrics were extracted at these depths using the "mri_vol2surf" function in FreeSurfer based on the transformation information from the co-registration between the non-diffusion-weighted image and the $T_1$-weighted image. Individual subject data were aligned to the FreeSurfer "fsaverage" cortical surface template using the FreeSurfer "mri_vol2surf" function. For visualization, we averaged the SANDI metrics in the template space across individuals scanned on each scanner.

### Histological atlas

Histological atlases, which display comprehensive detail on cortical cytoarchitecture and myeloarchitecture, served as references for our in vivo SANDI metrics. For cytoarchitecture, the BigBrain atlas was employed, which features a high-resolution 3D reconstruction of a human brain created from histological sections stained for cell bodies using the Merker technique[80,81]. The dataset provides a microscopic perspective on cell body density and its laminar patterns throughout the entire cortex. Myeloarchitectonic information was derived from the atlas developed by Nieuwenhuys et al.[116]. Based on the high-resolution mapping of cortical myelination patterns, this atlas presents the distribution and density of myelinated nerve fibers across different cortical areas and layers. To enable spatial correspondence with SANDI-derived microstructural metrics, we used cortical surface-based versions of the BigBrain cytoarchitectonic atlas (for intra-soma signal fraction, $f_{is}$) and the myeloarchitecture atlas (for intra-neurite signal fraction, $f_{in}$), both aligned to the FreeSurfer "fsaverage" template space[7,8].

### Statistics and reproducibility

All statistical analyses were performed using MATLAB (version 9.13, MathWorks, Natick, MA, USA). For the statistical analysis, the SANDI-derived microstructural metrics ($f_{is}$ and $f_{in}$) were extracted from each subject's native space. The Kolmogorov–Smirnov test was used to assess data normality. To compare SANDI metrics across scanners, two-sample t-tests were performed on values averaged across the full cortical depth of the entire cortex between the Connectome 2.0 and Connectome 1.0 scanners. To take advantage of the ultra-high-gradient strength of the Connectome 2.0 scanner, all subsequent analyses comparing SANDI metrics to histological atlases and prior cytoarchitectonic and myeloarchitectonic studies were performed exclusively using data from the Connectome 2.0 scanner. Paired t-tests were used to compare intra-soma signal fractions $f_{is}$ and intra-neurite signal fractions $f_{in}$ between the supragranular and infragranular layers. Pearson's correlation analysis was used to assess the correlation between intra-neurite signal fractions $f_{in}$ in the infragranular layer and myelin staining intensity from the myeloarchitecture atlas[8], using cortical labels defined by Nieuwenhuys' parcellation[116], excluding labels with missing or very low-intensity data. The infragranular layer was specifically examined because it contains a relatively high density of myelinated axons, which contribute significantly to the intra-neurite signal fraction $f_{in}$. Additional paired t-tests assessed regional differences in the intra-soma signal fractions $f_{is}$ between the motor and visual cortices across the full cortical depth (0–100%). For each subject, the median intra-soma signal fractions $f_{is}$ value within each region was used due to the relatively small size of the regions of interest. To examine the layer-specific relationship between cortical curvature and intra-soma signal fractions $f_{is}$, Pearson's correlation analyses were performed separately in the supragranular and infragranular layers, as well as at 10 and 90% cortical depths. The significance threshold was set at $P < 0.05$, corrected for multiple comparisons using FDR correction.

### Reporting summary

Further information on research design is available in the Nature Portfolio Reporting Summary linked to this article.

### Data availability

A Connectome 2.0 diffusion MRI dataset (raw and preprocessed DWIs) is publicly available on OpenNeuro (https://doi.org/10.18112/openneuro.ds006181.v1.0.0)[117]. Connectome 1.0 DWI datasets are publicly available via figshare (https://doi.org/10.6084/m9.figshare.c.5315474)[118]. All other data supporting the findings of this study are available from the corresponding author upon reasonable request, subject to applicable data-sharing agreements and conditions of reuse.

### Code availability

The code for diffusion MRI preprocessing and laminar analysis is publicly available on GitHub at https://github.com/Connectome20/diffusion_preproc_C2 and https://github.com/Connectome20/Laminar-SANDI.

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

## Acknowledgements

The data of BigBrain Merker staining intensity across cortical depths in Figs. 4, 5 and 6 are sourced from Paquola et al., *eLife* (2021). The data of myelin staining intensity across cortical depths and the histological image in Fig. 5 and Supplementary Fig. S8 are adapted from Foit et al., *NeuroImage* (2022), with permission from Elsevier. The cytoarchitectonic layer images in Fig. 6 are adapted from Campbell AW, *Histological Studies on the Localization of Cerebral Function* (1904), public domain. The schematic laminar cytoarchitecture image in Supplementary Fig. S9 is adapted from Fukutomi et al., *NeuroImage* (2019), licensed under the Creative Commons CC BY license. The histologically derived cell number profiles in Supplementary Fig. S11 are adapted from Hilgetag et al., *PLoS Computational Biology* (2006), licensed under the Creative Commons CC BY license. This study was supported by NIH under the award numbers: DP5OD031854, R01NS118187, P41EB015896, P41EB030006, U01EB026996, U24NS137077, R21AG085795, and R21AG082377. In addition, this research was supported by the Bio&Medical Technology Development Program of the National Research Foundation (NRF) funded by the Korean government (MSIT) (No. RS-2024-00411768) (H.L.). This study was further supported in part from the Hessen State Ministry of Higher Education, Research, and the Arts [grants "ADMIT" LOEWE/2/16/519/03/09.001(0001)/101 and LOEWE/4TP//519/05/02.002(0004)/107], NWO/ZonMw under the Rubicon award number 04520232330012 (K.S.C.), the German Research Foundation [grant: INST-169/22-1], and a postdoctoral fellowship from Huntington's Disease Society of America human biology project (X.Z.).

## Author contributions

H. Lee: conceptualization, formal analysis, investigation, writing–original draft, visualization. Y. Ma: conceptualization, data curation, investigation, writing–review and editing. K.S. Chan: data curation, investigation, methodology, writing–review and editing. E.A. Krijnen: conceptualization, formal analysis, investigation, methodology, writing–review and editing. L. Eskandarian: formal analysis, data curation. A. Bhatt: data curation, project administration. J. Gerold: data curation, project administration. M. Mahmutovic: resources. O. Puonti: methodology, writing–review and editing. X. Zeng: methodology. L.J. Deden Binder: methodology. B. Fischl: methodology, writing–review and editing. B. Keil: resources, writing–review and editing. G. Ramos-Llordén: investigation, writing–review and editing. E.C. Klawiter: data curation, investigation, writing–review and editing. H.H. Lee: conceptualization, methodology, investigation, project administration, writing–review and editing, funding acquisition. S.Y. Huang: conceptualization, methodology, investigation, project administration, resources, writing–review and editing, funding acquisition.

## Competing interests

The authors declare no competing interests.

## Additional information

**Supplementary information** The online version contains Supplementary material available at https://doi.org/10.1038/s42003-026-09887-2.

