## [Transparent Peer Review File · Communications Biology]

Visualizing cortical laminar architecture in the living human brain using next-generation ultra-high-gradient diffusion MRI

Corresponding Author: Dr Susie Huang

Version 0:

Reviewer comments:

Reviewer #1

(Remarks to the Author)

Thank you for inviting me to review this interesting manuscript, in which the authors present a novel dMRI approach for investigating cortical laminar architecture. The main findings include a peak in intracortical soma signal fraction, increasing intracortical neurite signal fraction, as well as some contrasts between cortical regions and associations with cortical folding. There is some evidence here that SANDI could be a powerful tool for in vivo histological/microstructural characterisation. However, the manuscript would benefit from additional clarifying images and more rigorous histological validation.

Major:

There are a number of uncertainties with the folding analysis.

To what extent could these relationships be driven by partial voluming? The limitation is noted in the discussion, but sulcal fundi are thin, especially in the deeper layers, whereas gyral crowns are thick with many more available voxels in the deeper layers. The difference between infragranular and supragranular (upper/lower layer) contrast in sulci is low. Could this be because the same voxels are being sampled for both in the sulci?

These folding results have no explicit histological support. Von Economo, BigBrain or the cited Hilgetag paper could be used to help establish the expected relationships based on cell body or staining density. Including this analysis in the figure 6 would strengthen the interpretation.

An image showing a fsoma for an example gyral crown and sulcal fundus would also help the viewer interpret this result. Equivolumetric effect - the 10% and 90% surfaces are likely to be sampling different layers in sulci and gyri due to the equivolume effect. This hasn't been modelled in the surface-based layering approach and could be another source of noise in this result.

M1 V1 analysis:

Figure 5 - It might be clearer to present this as regional profiles for vertices V1 vs M1, similar to the profile in Figure 3 which averages all cortical vertices. This would more clearly show the lower-layer divergence.

The histological backing for this lower-layer divergence finding is not very convincing. Why does the pial surface show an effect in dMRI, but not BigBrain?

Figure 5 comparing M1 and V1 would benefit from some example images from each region to show the reported effects.

In general, the manuscript would benefit from more dMRI images.

The main body currently contains no examples of the input fsoma or neurite images. It would be helpful to include at least one whole-brain example of each.

Methods into results: I appreciate this is due to the inverted paper format, but the results would benefit from some clarifications that are deep in the methods e.g:

The resolution of input and superresolved images would be included in the results and ideally in the abstract. The paper is about resolving laminar structure, and a key piece of information for this is the MRI resolution.

Segmentation of infra/supragranular - please clarify in the results that this is a deep learning approach separate to the surface-based approach.

More histological comparison:

Bigbrain staining is not a confirmed measure of cellular density, but a weak surrogate, and the Paquola et al., profiles additionally had some quantitative post-sampling corrections performed. The histological validations in the paper would be strengthened by the inclusion of von Economo gold standard histological estimates of layer and regional cellular density, which are available online.

Minor:

Figure 3. The discrepancy between 55% depth peak in soma density and 63% in the Merker could also be due to the differing methods for gm/wm boundary placement. The BigBrain surface is a histological/cell body contrast which doesn't necessarily align perfectly with the myelin driven MRI placement.

The authors note that Connectome 2 is "superior" to Connectome 1, but it is unclear which metric or experiment this refers to. The supplementary figure shows some differences, but no clear evaluation metric is provided to demonstrate in what sense 2.0 is better.

Reviewer #2

(Remarks to the Author)

This study covers novel applications of ultra high gradient dMRI scans using the Connectome MRI scanner for visualizing laminar patterns across the cortex.

The context in which this work is presented is currently lacking. The introduction does not adequately cover the existing studies and methods in the field of visualizing cortical laminar patterns using MRI. Many studies of cortical laminarity have been conducted over the last decade, using both IR and dMRI scan protocols, not only ex-vivo but also in-vivo.

Exploring the laminar structure of the cortex using IR MRI:

1. Lifshits S, Tomer O, Shamir I, et al. (2018). Resolution considerations in imaging of the cortical layers, *NeuroImage*, 164: 112-120.
2. Shamir I, et al. (2019). A Framework for Cortical Laminar Composition Analysis using Low Resolution T1 MRI Images, *Brain Structure and Function*, 224(4): 1457–1467.
3. Shamir I, Assaf Y (2021a). An MRI-based, data-driven model of cortical laminar connectivity, *Neuroinformatics*, vol. 19, 205–218.
4. Shamir I, Assaf Y (2021b). Modelling cortical laminar connectivity in the macaque brain, *Neuroinformatics*, 20(3): 559-573.
5. Lotan, E., Tomer, O., Tavor, I. et al. Widespread cortical dyslamination in epilepsy patients with malformations of cortical development. *Neuroradiology* 63, 225–234 (2021).
6. Shamir I, et al. (2022). Modelling the laminar connectome of the human brain, *Brain Structure and Function*, 227: 2153–2165.
7. Tomer O, Barazany D, Baratz Z, Tsarfaty G, Assaf Y. In vivo measurements of lamination patterns in the human cortex. *Hum Brain Mapp*. 2022 Jun 15;43(9):2861-2868.
9. Shamir I, et al. (2024). Clustering the cortical laminae: in vivo parcellation, *Brain Structure and Function*, 229(2): 443-458.

These papers appear in the review mentioned in the text (ref 11), which is misquoted in the introduction and referenced in an anatomical context only. Furthermore, the abovementioned references are all derived from a study referenced in the text (ref 19), which is also misquoted and referenced as an ex-vivo study, when it is in fact an in-vivo study on both rats and human subjects (the rats are compared to ex-vivo histology).

Exploring the laminar structure of the cortex using diffusion MRI:

10. Assaf, Y. Imaging laminar structures in the gray matter with diffusion MRI. *Neuroimage* 197, 677–688 (2019).
11. Shamir I, Assaf Y (2025). A guide to diffusion MRI and structural connectomics, *Nature Protocols*, 20: 317-335.

The first of the above two references specifically uses dMRI to map the laminarity of the cortex, and it is also discussed in the tutorial (the second reference).

Figure appearance and readability:

Figures 1-6 all include both black and white backgrounds and black and white text (or even colored/underlined text). I suggest using a white background for all FreeSurfer images within the larger images and only black text to improve appearance and overall readability.

On a more fundamental level, I do not understand the rationale behind the cohort selection. 21 subjects in one scanner, and 21 age and sex matched subjects in the other. Cortical laminarity does vary between subjects, not only based on age, sex, or even health. For example, it is assumed to play a role in cognitive ability and function, which varies between individuals. If the idea is to compare the laminar imaging ability of the two scanners, why not used the same group of subjects? A one-to-one comparison who minimize individual variabilities and better comparing the scanning ability.

On a conceptual level, the relation between the parameters modeled (intra-soma and intra-neurite fractions) and cyto- and myelo- architecture need to be better presented and discussed. Previous studies need to be mentioned and referenced here, starting from a link to myelarchitecture and progressing into cytoarchitecture.

Reviewer #3

(Remarks to the Author)

This study explores in vivo the cyto- and myeloarchitectonic organization of the human cerebral cortex using the Soma and Neurite Density Imaging (SANDI) model and high-gradient diffusion MRI (dMRI) (Connectome 2.0 scanner with 500 mT/m

gradient strength), enabling unprecedented in vivo microstructural imaging. Compared to the earlier Connectome 1.0 (300 mT/m), Connectome 2.0 offers higher resolution, shorter diffusion times, and improved sensitivity to cellular components (e.g., somas, neurites).

Cortical depth-dependent analysis of SANDI metrics provided a detailed characterization of cortical cytoarchitecture and myeloarchitecture in the living human brain, revealing distinct profiles that align with established histological patterns, particularly in the cell body density distribution and myelinated fibres across cortical layers. The results also suggest that Connectome 2.0 enhances microstructural specificity, particularly in detecting neurite density.

This work bridges postmortem histology and in vivo imaging, advancing non-invasive cortical microstructure characterization; it represents a significant methodological advancement in in vivo cortical microstructure imaging, with implications for basic neuroscience and clinical research.

The manuscript is very well written and contains all the relevant methodological details. The results are summarized and presented in a very informative way and the analysis is rigorous. I have only a few recommendations and comments for the authors:

- In the first paragraph of the Results, as well as in Figure 1; please mention and show schematically that dMRI data were super-resolved from 2 mm³ to 1mm³ isotropic resolution.

- Regarding the method for the super-resolution from Coupe et al NeuroImage 2013; it is unclear to me from the cited works and the results shown in the Supplementary Materials how accurate is the super resolution of high diffusion-weighted MRI images and how confident in the model prediction one can be (e.g., exclude hallucinations)? All the experiments and quantitative evaluations published are performed at relatively low b values and using simple signal representations (e.g., DTI). The authors should provide quantitative evidence of unbiased dMRI signal prediction and corresponding SANDI metrics when applying super-resolution to high b values. For example, one experiment could be to artificially down-sample the 2 mm³ dMRI to 4 mm³; add some extra Rician noise to match the original 2 mm³ SNR, and then apply the super-resolution method to predict the higher resolution, 2 mm³ dMRI. The existing ground truth at 2 mm³ can then be used to estimate, especially for the cortical voxels, mean squared error, mean absolute error and SSIM between predicted dMRI signal and SANDI maps and the ground truth ones.

- Regarding the results from Connectom 1.0, I have a couple of suggestions. 1) the authors could use the results on fis to argue more in favour of potential clinical translation, in the Discussion section: the dMRI protocol used on the Connectom 1.0 can be now easily mirrored on commercial clinical top-end scanners, such as the Siemens Cima X. As the clinical community will adopt these new technologies, replication studies and application to diseases will be feasible at scale. 2) The authors show that fis from Connectom 1.0 is not significantly different from Connectom 2.0, while fin it is. However, although fin estimates from Connectom 1.0 are significantly lower than those from Connectom 2.0, the ability of fin to capture myeloarchitectural characteristics across cortical layers may still be preserved. The authors should provide in supplementary Materials a similar analysis as in Figure 4, to show whether the impact of the different dMRI protocols between Connectom 1.0 and 2.0 is only a systematic shift of the values, which does not compromise differential analysis and sensitivity to myeloarchitecture.

- While I do understand why sharing "upon request" of the MRI data may be preferred; I am not in favour of not sharing openly the codes used to produce the results. I strongly encourage the authors to revise their Data availability statement to embrace a more open and reproducible approach by sharing all the codes on an open access platform, for example GitHub.

- The Discussion section is somewhat repetitive, particularly in reiterating key findings about Connectome 2.0's advantages, SANDI-derived metrics, and comparisons with histology. To improve readability without losing critical information, I suggest: merging scanner comparisons into a single, cohesive section; combining depth-dependent findings (e.g., layer-specific fis/fin trends) with histological validations and streamlining discussion of limitations by avoiding redundant mentions of model constraints/limitations.

Version 1:

Reviewer comments:

Reviewer #1

(Remarks to the Author)

Many thanks for the authors detailed responses to my suggestions.
I have no further comments to make.

Reviewer #2

(Remarks to the Author)

The manuscript has improved significantly in its conceptual flow, image readability, and data acquired and presented. All my concerns have been fully addressed. No further notes.

Reviewer #3

(Remarks to the Author)

The authors addressed all my concerns. I would now recommend acceptance of the paper.

Referee : 1

Thank you for inviting me to review this interesting manuscript, in which the authors present a novel dMRI approach for investigating cortical laminar architecture. The main findings include a peak in intracortical soma signal fraction, increasing intracortical neurite signal fraction, as well as some contrasts between cortical regions and associations with cortical folding.

There is some evidence here that SANDI could be a powerful tool for in vivo histological/microstructural characterisation. However, the manuscript would benefit from additional clarifying images and more rigorous histological validation.

Overall response: We thank the reviewer for the thoughtful and constructive evaluation of our manuscript. We appreciate the reviewer's insights regarding the methodological clarity and histological validation, and we have addressed each of the comments in the revised manuscript. Detailed responses to all points are provided below.

[Referee 1 Comment 1] There are a number of uncertainties with the folding analysis.

To what extent could these relationships be driven by partial voluming? The limitation is noted in the discussion, but sulcal fundi are thin, especially in the deeper layers, whereas gyral crowns are thick with many more available voxels in the deeper layers. The difference between infragranular and supragranular (upper/lower layer) contrast in sulci is low. Could this be because the same voxels are being sampled for both in the sulci?

Response: We agree with the reviewer that the folding-related analysis is limited by partial volume effects, particularly within thin sulcal fundi. Some voxels may be sampled by both supra- and infragranular layers in thin sulci, given that the cortical thickness is 1.5-3 mm, while the MRI data is sampled at 1 mm isotropic resolution after applying the super-resolution technique.

To address this concern, in addition to analyzing only two discrete layers (supragranular vs. infragranular), we performed a more detailed depth-dependent analysis by sampling across 21 equidistant cortical surfaces spanning the full cortical thickness (0–100%). This approach allowed us to assess the gradual, continuous pattern of microstructural variation across depths and to better characterize the divergent trends between supra- and infragranular layers. As shown in **Figure 6**, the results at 10% and 90% cortical depths more clearly differentiate gyri and sulci than the supra- and infragranular layers, and they clearly demonstrate opposite curvature-dependent relationships, consistent with the histological findings reported by Hilgetag et al. (2006) (see response to Referee 1, Comment 2 for more details).

Supplementary Figure S12. Relationship between intra-soma signal fraction f_{is} and cortical curvature with thickness ≥ 2 mm. Top: Cortical thickness map displayed on the fsaverage surface, with labels from the `aparc.a2009s.annot` parcellation. **Bottom:** Scatter plots show the relationships between curvature and f_{is} in supragranular and infragranular layers of cortical regions with thickness ≥ 2 mm, as well as at 10% and 90% cortical depths. CS: central sulcus; FIS: frontal inferior sulcus; PIS: parietal inferior sulcus; PostCG: postcentral gyrus; PreCG: precentral gyrus; TIS: temporal inferior sulcus.

In addition, we repeated the analysis by sampling vertices with cortical thickness ≥ 2 mm to minimize potential partial volume effects as shown in the figure above. The results remained consistent, confirming that the observed supra–infragranular differences and depth-dependent curvature effects are

not solely driven by partial volume effects within thin cortical regions. These supplementary results have been added as **Supplementary Figure S12**.

Updates in the revised manuscript

(P 17, Line 320) **These relationships were preserved when considering only vertices with cortical thickness ≥ 2 mm (Supplementary Figure S12).**

[Referee 1 Comment 2] **These folding results have no explicit histological support. Von Economo, BigBrain or the cited Hilgetag paper could be used to help establish the expected relationships based on cell body or staining density. Including this analysis in the figure 6 would strengthen the interpretation.**

Supplementary Figure S11. Curvature-dependent intra-soma signal fraction f_{is} and histological cell density distribution. Left: Scatter plots show the relationships between curvature and f_{is} at 10% and 90% cortical depths. **Right:** Histologically derived cell number profiles for different cortical folding patterns (adapted from Hilgetag et al., 2006⁷) show corresponding layer-specific trends.

Response: We appreciate the reviewer’s suggestion to provide explicit histological support for the observed folding-related intra-soma signal fraction f_{is} patterns. f_{is} was positively correlated with curvature in the supragranular layer (10% cortical depth), showing higher f_{is} in sulcal fundi compared with gyral crowns (two-sample t-test $P<0.001$). In contrast, f_{is} exhibited a negative correlation with curvature in deeper cortical regions (90% cortical depth), where higher f_{is} was observed in gyral regions (two-sample t-test $P=0.008$). These depth-dependent trends of f_{is} vs. curvature were consistent with the histological findings reported by Hilgetag et al. (2006) (**Supplementary Figure S11**, right panel),

which showed higher cell densities in sulcal supragranular layers (compared with gyral supragranular layer, gray bars) and higher densities in gyral infragranular layers (compared with sulcal infragranular layer, black bars).

Updates in the revised manuscript

(P 17, Line 317) In contrast, this relationship reversed in the deeper cortex: the infragranular layer showed a weak negative correlation ($r=-0.05$; $P=0.74$), and a stronger negative correlation emerged at 90% cortical depth ($r=-0.38$; $P=0.003$), indicating higher intra-soma signal fraction f_{is} in the gyral crowns than in the sulcal fundi at deeper cortical layers, consistent with previous histological observations⁸⁷ (Supplementary Figure S11).

[Referee 1 Comment 3] An image showing a fsoma for an example gyral crown and sulcal fundus would also help the viewer interpret this result.

Supplementary Figure S10. Representative intra-soma signal fraction f_{is} maps illustrating gyral and sulcal microstructural differences across cortical layers. f_{is} maps in a representative gyral/sulcal pair show the microstructural differences between gyral and sulcal regions within the infragranular layer (left) and the supragranular layer (right). These maps highlight the opposing gyral-sulcal relationships observed across cortical depths, with higher f_{is} values in the infragranular gyrus (compared with infragranular sulcus) and higher f_{is} values in the supragranular sulcus (compared with supragranular gyrus). Blue contours denote white-gray matter boundary (inner contour) and pial surface (outer contour), respectively, and the black contour indicates the boundary separating the supragranular and infragranular layers.

Response: We have added representative maps of intra-soma signal fraction f_{is} illustrating exemplary gyral crown and sulcal fundus findings. **Supplementary Figure S10** presents the enlarged f_{is} maps for both infragranular (left) and supragranular (right) examples, with the white-gray matter boundary (inner blue contour), the pial surface (outer blue contour), and the boundary separating supragranular and infragranular layers (black contour). These maps highlight the opposing gyral-sulcal relationships observed across cortical depths, with higher f_{is} values in the infragranular gyrus (compared with infragranular sulcus) and higher f_{is} values in the supragranular sulcus (compared with supragranular gyrus).

Updates in the revised manuscript

(P 17, Line 315) In the supragranular layer, a positive relationship was observed ($r=0.58$; $P<0.001$), indicating higher intra-soma signal fraction f_{is} in sulcal fundi compared to gyral crowns

(Supplementary Figure S10 for representative maps).

[Referee 1 Comment 4] Equivolumetric effect - the 10% and 90% surfaces are likely to be sampling different layers in sulci and gyri due to the equivolume effect. This hasn't been modelled in the surface-based layering approach and could be another source of noise in this result.

Supplementary Figure S7. Depth-dependent intra-soma signal fraction f_{is} profiles across the

whole cortex, derived using equivolumetric sampling (left) and equidistant sampling (right) implemented in LayNii toolbox.

Response: We thank the reviewer for this valuable comment. We agree that surface-based sampling does not fully capture the true anatomical layering of the cortex, since equidistant and equivolumetric surfaces correspond to slightly different cytoarchitectonic layers in gyri and sulci. In our analysis, we used the standard “vol2surf” pipeline in FreeSurfer with equidistant sampling. To assess the robustness, we additionally ran the analysis using “LayNii” laminar-fMRI toolbox with both equidistant and equivolumetric schemes and found similar depth-dependent patterns of intra-soma signal fraction f_{is} for the two surface-based sampling methods. Both sampling schemes showed similar depth-dependent patterns of the intra-soma signal fraction f_{is} , each peaking at ~55% cortical depth (**Supplementary Figure S7**). Nevertheless, we acknowledge that a more anatomically precise segmentation of cortical layers at the voxel/vertex level is needed.

In addition, we have updated the Introduction to include MRI-based laminar imaging approaches, particularly T_1 -based methods that identify six distinct components corresponding to cortical layers (see response to Referee 2, Comment 1 for more details). We have incorporated this consideration into the revised Discussion, noting that such MRI-based laminar segmentation, if applied to SANDI-derived metrics, could provide a more direct correspondence with laminar cytoarchitecture and myeloarchitecture seen on histology.

Updates in the revised manuscript

(P 14, Line 257) Additionally, both equidistant and equivolumetric sampling schemes in the LayNii laminar-fMRI toolbox yielded a peak in the intra-soma signal fraction f_{is} at the same depth (~55%) (**Supplementary Figure S7**).

(P 23, Line 448) In parallel, advances in MRI-based layer segmentation are expected to strengthen further the translational potential of the laminar SANDI framework^{11,37}. Incorporating later-specific T_1 -

based segmentation could enable more accurate assignment of SANDI-derived microstructural metrics to their corresponding cortical layers⁴⁶ and provide a more direct *in vivo* surrogate for cytoarchitectonic and myeloarchitectonic organization.

[Referee 1 Comment 5] M1 V1 analysis:

Figure 5 - It might be clearer to present this as regional profiles for vertices V1 vs M1, similar to the profile in Figure 3 which averages all cortical vertices. This would more clearly show the lower-layer divergence.

Figure 6. Comparison of SANDI-derived intra-soma signal fraction f_{is} between the motor cortex and visual cortex across cortical depths, measured using the Connectome 2.0 scanner. Top: Representative intra-soma signal fraction f_{is} maps from motor and visual cortices are shown alongside corresponding cytoarchitectonic layer profiles from Kandel et al., 2013⁸³, with permission from McGraw-Hill. White arrows indicate the low f_{is} values near the pial surface, while black arrows

highlight the mid-depth peak observed in the motor cortex. **Bottom:** Intra-soma signal fraction f_{is} , alongside Merker staining data from the BigBrain atlas for cytoarchitecture, across the cortical depths. Statistically significant differences are indicated (*: FDR- $P < 0.05$). The data of BigBrain Merker staining intensity across cortical depths are sourced from Paquola et al., 2021⁷.

Response: We appreciate the reviewer's suggestion to clarify the regional depth profiles for the motor and visual cortices. In **Figure 6**, we have now included the profiles of the intra-soma signal f_{is} and BigBrain cell body staining intensity across cortical depth (0-100%) for the two cortical regions separately, along with a comparison of f_{is} between the the motor and visual cortices. This representation more clearly illustrates the laminar-specific differences, showing greater f_{is} and BigBrain cell body staining intensity in the deeper cortical layers of the visual cortex compared with the motor cortex. Furthermore, f_{is} peak appeared at a shallower depth in the motor cortex (~40%) than in the visual cortex (~65%). A similar pattern was observed in the BigBrain staining profiles (~53% depth in motor cortex and ~67% depth in visual cortex).

Furthermore, in response to Referee 1, Comment 7, we have added representative maps of the intra-soma signal fraction f_{is} alongside the corresponding cytoarchitectonic layer profiles in **Figure 6**. White arrows indicate regions near the pial surface with low f_{is} values, whereas black arrows highlight the shallower peak observed in the motor cortex.

Updates in the revised manuscript

(P 15, Line 288) Specifically, the intra-soma signal fraction f_{is} was significantly higher in the visual cortex at 0% (FDR- $P=0.002$) and across the 65-100% depths (FDR- $P<0.009$), a pattern aligned with both cytoarchitectonic layer profiles⁸⁵ and Merker staining intensity from the BigBrain atlas. In addition to differences in magnitude, the two regions differed in the depth at which the intra-soma signal fraction f_{is} peaked. The motor cortex showed a shallower peak at ~40% depth (**Figure 6**), whereas the visual cortex exhibited a deeper peak at ~65%. This depth shift is reflected in the BigBrain cell body staining intensity profiles, which show a peak at ~53% depth in the motor cortex and ~67% in the visual cortex.

[Referee 1 Comment 6] The histological backing for this lower-layer divergence finding is not very convincing. Why does the pial surface show an effect in dMRI, but not BigBrain?

Response: We appreciate the reviewer's insightful comment. We agree that the divergence observed near the pial surface in the SANDI-derived f_{is} maps is not directly reproduced in the BigBrain cell body staining intensity profiles. This superficial-layer discrepancy is most likely driven by modality-specific limitations. In MRI, cortical surfaces generated at 1 mm isotropic resolution are susceptible to partial-volume effects from CSF. As noted by the Referee 1 in Comment 12, differences in gray-white matter boundary and pial surface placement between MRI and histology can further introduce depth shifts in the sampled profiles. These effects are most pronounced near the pial surface and can lead to local deviations that are not present in the high-resolution (20 μm) BigBrain dataset. To clarify this point, we have provided the profiles of the intra-soma signal f_{is} across cortical depth (0-100%) along with the representative maps of the intra-soma signal fraction f_{is} in **Figure 6**. Additionally, to mitigate these sources of bias, all analyses were performed across the full cortical depth range (0-100%) at 5% intervals. We have revised the Discussion accordingly.

Updates in the revised manuscript

(P 24, Line 466) Additionally, discrepancies between the *in vivo* MRI surfaces and the histological surfaces can introduce depth-dependent misalignment. In the BigBrain atlas, the gray-white matter boundary is delineated from cell body staining contrast, whereas FreeSurfer MRI surfaces rely on T_1 -weighted intensity, which is primarily driven by myelin. These difference in boundary definition likely contributes to the observed 55% versus 63% peak depth discrepancy between SANDI-derived intra-soma signal fraction f_{is} and BigBrain Merker cell density profiles. Sampling near the pial surface and the gray-white matter boundary is further complicated by partial volume effects from CSF, local curvature differences, and the resolution gap between ultra-high-resolution (20 μm) BigBrain data and

[Referee 1 Comment 7] Figure 5 comparing M1 and V1 would benefit from some example images from each region to show the reported effects.

Response: We thank the reviewer for this suggestion. In the revised manuscript, we have now included representative maps of the intra-soma signal fraction f_{is} for both the motor and visual cortices in the updated Figure 6. (see response to Referee 1, Comment 5 for more details).

[Referee 1 Comment 8] In general, the manuscript would benefit from more dMRI images. The main body currently contains no examples of the input fsoma or neurite images. It would be helpful to include at least one whole-brain example of each.

Figure 1. Representative SANDI-derived microstructural metrics, group-averaged across 21 subjects and registered to MNI space.

Response: We appreciate the reviewer's suggestion to include more representative diffusion MRI images. In response, we have added whole-brain examples of all SANDI-derived metrics, averaged across 21 subjects and registered to MNI space, in the revised Figure 1. In addition, Figure 2 has been updated to include example diffusion-weighted images (2 mm and super-resolved 1 mm) to more clearly illustrate the input data used for SANDI fitting and the processing pipeline (see response to Referee 1, Comment 9 for more details).

Updates in the revised manuscript

(P 10, Line 179) The SANDI model provided parametric maps of six microstructural metrics: intra-soma signal fraction (f_{is}), intra-neurite signal fraction (f_{in}), extracellular signal fraction ($f_{ec}=1 - f_{is} - f_{in}$), apparent soma radius (R_s), intra-neurite diffusivity (D_{in}), and extracellular diffusivity (D_{ec}) (Figure 1).

[Referee 1 Comment 9] Methods into results: I appreciate this is due to the inverted paper format, but the results would benefit from some clarifications that are deep in the methods e.g: The resolution of input and superresolved images would be included in the results and ideally in the abstract. The paper is about resolving laminar structure, and a key piece of information for this is the MRI resolution.

Figure 2. Framework for cortical depth-dependent microstructure analysis of SANDI metrics, with comparisons to histological atlases.

Response: Thank you for this valuable comment. In the revised manuscript, we have now clarified the resolution details by updating Figure 2 to illustrate the super-resolution image processing step (from 2 mm to 1 mm diffusion MRI) and by adding this information to both the abstract and the second

paragraph of the Results section describing the analysis pipeline.

Updates in the revised manuscript

(P 3, Line 42) Leveraging the next-generation Connectome MRI scanner (maximum gradient strength=500mT/m, slew rate=600T/m/s), we characterized *in vivo* cortical laminar cytoarchitecture and myeloarchitecture through cortical depth-dependent analyses of soma and neurite density imaging (SANDI) metrics derived from **1 mm** diffusion MRI generated using a super-resolution technique.

(P 10, Line 187) To assess cortical microstructural features in the living human brain and to relate these depth-dependent patterns to cytoarchitectonic and myeloarchitectonic histological atlases, we applied a framework using a multi-step analysis pipeline (**Figure 2**) that includes anatomical parcellation, surface reconstruction, **super-resolution image processing of the diffusion-weighted images (2 mm to 1 mm isotropic; Supplementary Figure S1) prior to SANDI fitting**, deep learning-based supragranular/infragranular segmentation, and laminar sampling of SANDI metrics (intra-soma signal fraction f_{is} and intra-neurite signal fraction f_{in}) using FreeSurfer's "mri_vol2surf".

[Referee 1 Comment 10] Segmentation of infra/supragranular - please clarify in the results that this is a deep learning approach separate to the surface-based approach.

Response: We have clarified in the revised Results section that the segmentation of supragranular and infragranular layers was performed using a deep learning-based approach.

Updates in the revised manuscript

(P 10, Line 187) To assess cortical microstructural features in the living human brain and to relate these depth-dependent patterns to cytoarchitectonic and myeloarchitectonic histological atlases, we applied a

framework using a multi-step analysis pipeline (**Figure 2**) that includes anatomical parcellation, surface reconstruction, super-resolution image processing of the diffusion-weighted images (2 mm to 1 mm isotropic; **Supplementary Figure S1**) prior to SANDI fitting, **deep learning-based supragranular/infragranular segmentation**, and laminar sampling of SANDI metrics (intra-soma signal fraction f_{is} and intra-neurite signal fraction f_{in}) using FreeSurfer's "mri_vol2surf".

[Referee 1 Comment 11] More histological comparison:

Bigbrain staining is not a confirmed measure of cellular density, but a weak surrogate, and the Paquola et al., profiles additionally had some quantitative post-sampling corrections performed. The histological validations in the paper would be strengthened by the inclusion of von Economo gold standard histological estimates of layer and regional cellular density, which are available online.

Supplementary Figure S9. Depth-dependent SANDI-derived intra-soma signal fraction f_{is} across von Economo cortical types. Left: von Economo's cytoarchitectonic cortical type parcellation displayed on the fsaverage cortical surface, alongside schematic laminar cytoarchitecture for each cortex type (adapted from Fukutomi et al., 2019⁶, with permission from Nature Portfolio). **Right:** Depth-dependent profiles of the SANDI-derived intra-soma signal fraction f_{is} and BigBrain Merker staining intensity for each cortical type.

Response: We thank the reviewer for this helpful suggestion. Following the comment, we incorporated the von Economo's cytoarchitectonic cortical type classification (agranular, frontal, parietal, polar, and granular) using the ENIGMA toolbox to provide stronger histological support for the laminar patterns observed in the SANDI-derived intra-soma signal fraction f_{is} profiles. As shown in newly added **Supplementary Figure S9**, distinct depth-dependent f_{is} profiles were observed across von Economo's cytoarchitectonic classes. Specifically, agranular and frontal cortices show relatively shallow peak (~45%), consistent with their diminished or absent layer IV and the predominance of larger pyramidal neurons in layers III and V. In contrast, granular cortex exhibits deeper peak (~65% depth), reflecting its well-developed layer IV, which contains a dense population of small granule cells, consistent with the classical laminar architecture in von Economo's histological atlas.

Updates in the revised manuscript

(P 16, Line 307) The intra-soma signal fraction f_{is} profiles across von Economo's cytoarchitectonic cortical types revealed depth-dependent patterns unique to each cortical class (agranular, frontal, parietal, polar, and granular) (**Supplementary Figure S9**). Agranular and frontal cortices exhibited peak values at relatively shallow cortical depth (~45%), whereas granular cortex presented a deeper peak (~65%), reflecting the prominent layer IV in the granular cortex, as described in the laminar cytoarchitectonic features of von Economo's histological atlas⁸⁶.

(P 21, Line 391) To provide broader histological context for these region- and depth-dependent patterns, we additionally examined intra-soma signal fraction f_{is} profiles across von Economo's cytoarchitectonic cortical types⁸⁶. Distinct laminar profiles emerged across these types. Agranular cortices (including the primary motor cortex) and frontal cortices showed relatively shallow peaks (~45% depth), reflecting their diminished or absent layer IV and the predominance of larger pyramidal neurons in layer III (and layer V). In contrast, the deeper peaks (~65% depth) observed in granular cortices (including primary visual, somatosensory, and auditory cortices) align with their well-developed layer IV, which contains

a dense population of small granule cells that markedly contribute to the soma signal fraction f_{is} , aligning closely with the classical laminar architecture described in von Economo's histological atlas.

(P 28, Line 571) To further characterize the laminar profiles with region-specific cytoarchitectonic information, cortical regions were additionally classified according to von Economo's cortical types using the ENIGMA toolbox¹¹⁴, which provides surface-based parcellations of the agranular, frontal, parietal, polar, and granular classes on the FreeSurfer "fsaverage" cortical surface template.

[Referee 1 Comment 12] Minor:

Figure 3. The discrepancy between 55% depth peak in soma density and 63% in the Merker could also be due to the differing methods for gm/wm boundary placement. The BigBrain surface is a histological/cell body contrast which doesn't necessarily align perfectly with the myelin driven MRI placement.

Response: We thank the reviewer for this insightful comment and fully agree. The slight difference between the peak depths (55% vs. 63%) may indeed arise from the different definitions of the gray/white matter boundary in FreeSurfer-derived MRI surfaces and the BigBrain histological surfaces. The FreeSurfer boundary is defined using T_1 -weighted MRI intensity contrast, which is primarily driven by myelin, whereas the BigBrain boundary is delineated from histological cell body (cytoarchitectonic) contrast obtained from Merker staining. These differences could result in a small misalignment between the two surfaces. We have added this clarification to the Discussion section.

Updates in the revised manuscript

(P 24, Line 466) Additionally, discrepancies between the *in vivo* MRI surfaces and the histological surfaces can introduce depth-dependent misalignment. In the BigBrain atlas, the gray-white matter boundary is delineated from cell body staining contrast, whereas FreeSurfer MRI surfaces rely on T_1 -

weighted intensity, which is primarily driven by myelin. These difference in boundary definitions likely contributes to the observed 55% versus 63% peak depth discrepancy between SANDI-derived intra-soma signal fraction f_{is} and BigBrain Merker cell density profiles. Sampling near the pial surface and the gray-white matter boundary is further complicated by partial volume effects from CSF, local curvature differences, and the resolution gap between ultra-high-resolution (20 μm) BigBrain data and 1 mm *in vivo* dMRI.

[Referee 1 Comment 13] The authors note that Connectome 2 is “superior” to Connectome 1, but it is unclear which metric or experiment this refers to. The supplementary figure shows some differences, but no clear evaluation metric is provided to demonstrate in what sense 2.0 is better.

Supplementary Figure S5. SANDI-derived microstructural metrics on Connectome 2.0 (C2.0) and Connectome 1.0 (C1.0) scanners. Left: Cortical maps of the intra-soma signal fraction f_{is} and intra-neurite signal fraction f_{in} derived from SANDI, averaged across 21 individuals. The third column shows the P -values of vertex-wise differences in these metrics between Connectome 2.0 and Connectome 1.0, indicating regions with statistically significant differences. **Right:** Boxplots summarizing intra-soma signal fraction f_{is} and intra-neurite signal fraction f_{in} across individuals ($N=21$ for both scanners), including a subset of participants who were scanned on both scanners ($N=3$), with statistically significant differences of f_{in} (*: $\text{FDR-}P < 0.05$) between Connectome 2.0 and Connectome 1.0 scanners. The box represents the 1.96 standard error of the mean (95% confidence interval), and the line represents the 1 standard deviation.

Response: We thank the reviewer for this comment. To clarify in what sense Connectome 2.0 is superior to Connectome 1.0, we provide detailed quantitative evidence demonstrating its enhanced microstructural sensitivity. The higher intra-neurite signal fraction f_{in} values observed with the Connectome 2.0 scanner (compared with Connectome 1.0) in **Supplementary Figure S5** likely reflect

the benefits of shorter diffusion time and echo time (TE), which together minimize inter-compartmental water exchange and improve the overall signal-to-noise ratio (SNR), thereby enhancing sensitivity to f_{in} . Additionally, our recent work using the Connectome 2.0 scanner has demonstrated superior sensitivity and microstructural contrast for soma-related SANDI metrics compared with Connectome 1.0 (Ramos-Llordén et al., *Nature Biomedical Engineering*, 2025). In particular, the intra-soma signal fraction f_{is} revealed greater regional contrast across Brodmann areas (4, 3a, 3b, and 1), together with smaller inter-subject variability on Connectome 2.0, closely matching prior histological findings. These distinctions were less pronounced when estimated on Connectome 1.0. Moreover, Connectome 2.0 showed increased sensitivity to smaller soma populations. Together, these results highlight that Connectome 2.0 provides superior microstructural sensitivity due to its stronger gradient strength and shorter diffusion times.

Updates in the revised manuscript

(P 19, Line 353) **Enhanced microstructural sensitivity with Connectome 2.0**

The Connectome 2.0 scanner shows enhanced microstructural sensitivity relative to Connectome 1.0, enabling more precise characterization of both neurite and soma compartments. The stronger gradient system allows for substantially shorter diffusion times and echo times, which reduce inter-compartmental water exchange effects^{74,92-94} that are not modeled in SANDI⁷⁸ and improve SNR, resulting in higher intra-neurite signal fraction f_{in} values. Beyond neurites, although whole-cortex averages of the intra-soma signal fraction f_{is} did not differ between scanners, more detailed subregional analyses using Connectome 2.0 revealed improved performance for soma-related SANDI metrics. Specifically, the intra-soma signal fraction f_{is} exhibited greater regional contrast across Brodmann areas (BA4, 3a, 3b, and 1), closely matching prior histological findings, and reduced inter-subject variability compared with Connectome 1.0³⁰. In addition, Connectome 2.0 shows greater sensitivity to smaller soma populations.

Referee : 2

This study covers novel applications of ultra high gradient dMRI scans using the Connectome MRI scanner for visualizing laminar patterns across the cortex.

The context in which this work is presented is currently lacking. The introduction does not adequately cover the existing studies and methods in the field of visualizing cortical laminar patterns using MRI. Many studies of cortical laminarity have been conducted over the last decade, using both IR and dMRI scan protocols, not only ex-vivo but also in-vivo.

Overall response: We thank the reviewer for highlighting the need for stronger contextualization in the manuscript, particularly in the Introduction. In response, we have revised the manuscript to more comprehensively describe prior work on cortical laminar imaging using inversion recovery (IR)- and diffusion MRI (dMRI)-based approaches and have improved the figure presentation as suggested. Detailed responses are provided below.

[Referee 2 Comment 1] Exploring the laminar structure of the cortex using IR MRI:

- 1. Lifshits S, Tomer O, Shamir I, et al. (2018). Resolution considerations in imaging of the cortical layers, *NeuroImage*, 164: 112-120.**
- 2. Shamir I, et al. (2019). A Framework for Cortical Laminar Composition Analysis using Low Resolution T1 MRI Images, *Brain Structure and Function*, 224(4): 1457–1467.**
- 3. Shamir I, Assaf Y (2021a). An MRI-based, data-driven model of cortical laminar connectivity, *Neuroinformatics*, vol. 19, 205–218.**
- 4. Shamir I, Assaf Y (2021b). Modelling cortical laminar connectivity in the macaque brain, *Neuroinformatics*, 20(3): 559-573.**

5. Lotan, E., Tomer, O., Tavor, I. et al. Widespread cortical dyslamination in epilepsy patients with malformations of cortical development. *Neuroradiology* 63, 225–234 (2021).
6. Shamir I, et al. (2022). Modelling the laminar connectome of the human brain, *Brain Structure and Function*, 227: 2153–2165.
7. Tomer O, Barazany D, Baratz Z, Tsarfaty G, Assaf Y. In vivo measurements of lamination patterns in the human cortex. *Hum Brain Mapp.* 2022 Jun 15;43(9):2861-2868.
9. Shamir I, et al. (2024). Clustering the cortical laminae: in vivo parcellation, *Brain Structure and Function*, 229(2): 443-458.

These papers appear in the review mentioned in the text (ref 11), which is misquoted in the introduction and referenced in an anatomical context only. Furthermore, the abovementioned references are all derived from a study referenced in the text (ref 19), which is also misquoted and referenced as an ex-vivo study, when it is in fact an in-vivo study on both rats and human subjects (the rats are compared to ex-vivo histology).

Exploring the laminar structure of the cortex using diffusion MRI:

10. Assaf, Y. Imaging laminar structures in the gray matter with diffusion MRI. *Neuroimage* 197, 677–688 (2019).
11. Shamir I, Assaf Y (2025). A guide to diffusion MRI and structural connectomics, *Nature Protocols*, 20: 317-335.

The first of the above two references specifically uses dMRI to map the laminarity of the cortex, and it is also discussed in the tutorial (the second reference).

Response: We thank the reviewer for pointing out the extensive body of work on cortical laminar imaging using IR and dMRI techniques. We agree that our initial introduction did not sufficiently cite or appropriately contextualize these foundational studies. Accordingly, we have substantially revised the introduction to provide a more comprehensive overview of existing MRI-based approaches to characterize cortical lamination. Specifically:

1. **Inversion recovery T_1 -based approaches:** We now describe the extensive work on quantitative T_1 mapping for cortical laminar analysis (ref. 38-52, including those suggested by the reviewer), encompassing frameworks for laminar composition analysis, laminar connectivity modeling, clinical applications in detecting cortical dyslamination in epilepsy, and recent efforts toward *in vivo* laminar parcellation. Additionally, in response to Referee 2, Comment 4, we clarify that these T_1 -derived laminar profiles primarily reflect myeloarchitectonic features and provide an indirect measure of cytoarchitecture.
2. **Diffusion MRI approaches:** We have incorporated dMRI-based studies for mapping cortical laminae (refs. 53-57) and emphasized that diffusion-based approaches such as FA, RD, and fODF show depth-dependent variations and characteristic orientation patterns of cortical microstructure. We highlight that dMRI combined with advanced biophysical modeling, such as the SANDI model, offer complementary microstructural contrast beyond T_1 -based approaches by capturing both cyto- and myeloarchitectonic features. We now describe how the SANDI model leverages diffusion signals from the soma and neurite compartments.
3. **Correction of misquotations:** We have corrected the previously misquoted references and now clearly cite the studies including *ex vivo* MRI for cortical lamination.

We believe these updates strengthen the contextual foundation of the manuscript, properly acknowledge prior contributions in the field, and clarify how our high-gradient SANDI framework offers a distinct and complementary advance toward noninvasive mapping of human cortical lamination.

Updates in the revised manuscript

(P 6, Line 100) Over the decades, substantial progress has been made toward non-invasively visualizing cortical laminar patterns in the living human brain using advanced MRI methods^{11,37}. These efforts utilize different contrast mechanisms (T_1 , T_2 , T_2^* , phase, susceptibility, magnetization transfer, and dMRI) to study cortical gray matter organization on the laminar level³⁸⁻⁴². Among these methods, T_1 -weighted approaches have proven to be the most studied and demonstrated to be robust and practical,

with early studies showing that myelination shortens T_1 values and that T_1 -weighted images contain laminar signatures reflecting the underlying myeloarchitecture, revealing six T_1 -defined components that correspond to histological layers⁴³⁻⁴⁵. Subsequent studies extended this framework by leveraging quantitative T_1 inversion-recovery MRI protocols for whole-brain laminar mapping while addressing partial volume effects^{46,47}, modeling cortical laminar connectivity⁴⁸⁻⁵⁰, and applying laminar imaging in clinical populations, such as detecting dyslamination in epilepsy⁵¹. Although T_1 -derived laminar patterns have recently been shown to correlate with cytoarchitectonic regions⁵², T_1 relaxation primarily reflects myeloarchitecture rather than cytoarchitecture and therefore provides an indirect measure of cellular organization¹¹.

(P 6, Line 114) Beyond these T_1 -based approaches, dMRI provides a distinct and complementary source of microstructural contrast that is sensitive to complex features such as fiber orientation, cellular density, and other microscopic tissue properties³⁴⁻³⁶. Increasing evidence suggests that the cortical diffusion signal is layer-dependent and reveals microstructural features that are complementary to conventional T_1 -weighted or quantitative relaxometry methods^{53,54}. Even basic diffusion tensor imaging (DTI) shows depth-dependent variations in fractional anisotropy and radial diffusion within the cortex^{53,55,56}, while the fiber orientation distribution function (fODF) exhibits robust radial and tangential patterns that vary with cortical curvature and laminar architecture^{53,56,57}. These findings underscore the strong potential of dMRI for providing a more comprehensive characterization of cortical cyto- and myeloarchitecture *in vivo*. Such laminar sensitivity can be further supported by advanced biophysical modeling that disentangles the heterogeneous cellular contributors to the diffusion signal and improves specificity to compartment-level features reflecting the underlying laminar organization³⁵.

(P 8, Line 153) Soma and neurite compartments reflect distinct biological properties related to cortical architecture, where intra-neurite fraction is sensitive to axonal and dendritic organization associated with myeloarchitecture, complementing established myelin-sensitive contrasts. The intra-soma fraction provides additional sensitivity to cell body density and captures cytoarchitectonic features, thereby

offering clear advantages over existing dMRI contrasts^{76,79}.

(P 23, Line 446) Since cortical fODFs exhibit distinct radial and tangential organization across laminae⁵³, integrating fODF-based orientation information with SANDI-derived compartment metrics may enhance the laminar sensitivity of dMRI-based cortical characterization. In parallel, advances in MRI-based layer segmentation are expected to strengthen further the translational potential of the laminar SANDI framework^{11,37}. Incorporating layer-specific T_1 -based segmentation could enable more accurate assignment of SANDI-derived microstructural metrics to their corresponding cortical layers⁴⁶ and provide a more direct *in vivo* surrogate for cytoarchitectonic and myeloarchitectonic organization.

[Referee 2 Comment 2] Figure appearance and readability:

Figures 1-6 all include both black and white backgrounds and black and white text (or even colored/underlined text). I suggest using a white background for all FreeSurfer images within the larger images and only black text to improve appearance and overall readability

Figure 4. SANDI-derived intra-soma signal fraction f_{is} on the Connectome 2.0 scanner, alongside Merker staining data from the BigBrain atlas for cytoarchitecture across the cortical depths. The data of BigBrain Merker staining intensity across cortical depths are sourced from Paquola et al., 2021⁷.

Response: Thank you for the suggestion. We have revised the figures to improve visual clarity. All FreeSurfer images now use a white background, and all text annotations have been changed to black

font for improved readability. Here we show the revised **Figure 4** as an example of the update.

[Referee 2 Comment 3] On a more fundamental level, I do not understand the rationale behind the cohort selection. 21 subjects in one scanner, and 21 age and sex matched subjects in the other. Cortical laminarity does vary between subjects, not only based on age, sex, or even health. For example, it is assumed to play a role in cognitive ability and function, which varies between individuals.

If the idea is to compare the laminar imaging ability of the two scanners, why not used the same group of subjects? A one-to-one comparison who minimize individual variabilities and better comparing the scanning ability.

Supplementary Figure S5. SANDI-derived microstructural metrics on Connectome 2.0 (C2.0) and Connectome 1.0 (C1.0) scanners. Left: Cortical maps of the intra-soma signal fraction f_{is} and intra-neurite signal fraction f_{in} derived from SANDI, averaged across 21 individuals. The third column shows the P -values of vertex-wise differences in these metrics between Connectome 2.0 and Connectome 1.0, indicating regions with statistically significant differences. Right: Boxplots summarizing intra-soma signal fraction f_{is} and intra-neurite signal fraction f_{in} across individuals ($N=21$ for both scanners), including a subset of participants who were scanned on both scanners ($N=3$), with statistically significant differences of f_{in} (*: FDR- $P<0.05$) between Connectome 2.0 and Connectome 1.0 scanners. The box represents the 1.96 standard error of the mean (95% confidence interval), and the line represents the 1 standard deviation.

Response: We agree that scanning the same participants on both scanners would have minimized inter-individual variability and provided a more direct comparison of laminar imaging performance. However, the Connectome 1.0 scanner was decommissioned in mid-2023 and the Connectome 2.0 scanner was subsequently installed in the same scanning room. A substantial portion of the data acquired on the Connectome 1.0 scanner had been obtained before the COVID-19 pandemic, and many of these individuals were lost to follow-up and/or declined to return for scans upon installation of the new

scanner. As a result, it was not feasible to acquire more data from the same group of individuals on both systems. Nevertheless, the reviewer's comment prompted us to examine the small subset of individuals ($N = 3$) who were scanned on both scanners (**Supplementary Figure S5**) with inter-scan intervals of 4, 5, and 11 months. For the remaining participants, we recruited two separate cohorts of healthy young adults (aged 19-40 years) for each scanner, matched for age and sex, and acquired data under similar imaging protocols to minimize group-level differences. The three participants who underwent on both systems showed the same trends as those observed in the overall sample, with significantly higher intra-neurite signal fraction f_{in} values on Connectome 2.0 (FDR- $P=0.02$) and no significant differences in intra-soma signal fraction f_{is} (FDR- $P=0.77$).

Updates in the revised manuscript

(P 11, Line 199) A total of 39 healthy adults under 40 years of age participated in this study at Massachusetts General Hospital. **Three participants completed scans on both the Connectome 2.0 MRI scanner (G_{\max} of 500 mT/m and maximum slew rate of 600 T/m/s) and the Connectome 1.0 MRI scanner (G_{\max} of 300 mT/m and maximum slew rate of 200 T/m/s).** The Connectome 2.0 cohort consisted of 21 individuals (14 females, 7 males; mean age: 29.0 ± 4.5 years; age range: 19-37) who underwent MRI scans on the newly installed 3T Connectome 2.0 system. The Connectome 1.0 cohort included 21 age- and sex-matched participants (14 females, 7 males; mean age: 28.7 ± 6.2 years; age range: 19-40) who were scanned on the 3T Connectome 1.0 system. **The three participants who completed scans on both systems were females aged 27, 32, and 36 years.**

(P 12, Line 220) **The three participants who underwent on both systems showed the same trends as those observed in the overall sample, with significantly higher intra-neurite signal fraction f_{in} values on Connectome 2.0 (FDR- $P=0.02$) and no significant differences in intra-soma signal fraction f_{is} (FDR- $P=0.77$).**

(P 26, Line 507) A subset of participants (N=3) were recruited for both scanners within inter-scan intervals of 4-11 months.

[Referee 2 Comment 4] On a conceptual level, the relation between the parameters modeled (intra-soma and intra-neurite fractions) and cyto- and myelo- architecture need to be better presented and discussed. Previous studies need to be mentioned and referenced here, starting from a link to myelarchitecture and progressing into cytoarchitecture.

Response: We appreciate the reviewer's insightful comment. In the revised manuscript, we have updated the Introduction to clarify the conceptual relationship between the SANDI-derived parameters and the underlying cyto- and myeloarchitecture as suggested.

First, we have expanded the Introduction to summarize previous MRI-based laminar imaging studies showing that T_1 -weighted and quantitative T_1 methods primarily reflect myeloarchitectonic features, including known laminar myelin patterns, while recent work has demonstrated associations with cytoarchitectonic patterns as well (Shamir et al., *Brain Structure and Function*, 2024).

Second, we now describe how dMRI provides complementary microstructural contrast sensitive to fiber orientation, cellular density, and other microscopic tissue properties. We highlight accumulating evidence that the cortical diffusion signal is layer-dependent, supporting its relevance for characterizing both cyto- and myeloarchitecture.

Third, we articulate how SANDI-derived metrics relate to these biological substrates: the intra-neurite fraction captures axonal and dendritic organization associated with myeloarchitecture, while the intra-soma fraction reflects cell body density reflecting cytoarchitecture.

In addition, we have updated the Clinical translation & Future directions section of the Discussion to highlight that fODF-based orientation information and T_1 -based layer segmentation could complement and further enhance the laminar specificity of SANDI-derived metrics.

These revisions provide a conceptual progression from myeloarchitectonic contrast in T_1 -

based methods to cytoarchitectonic sensitivity in diffusion-based approaches and improve the interpretability of SANDI metrics with respect to established histological features and previously published studies.

Updates in the revised manuscript

(P 6, Line 100) Over the decades, substantial progress has been made toward non-invasively visualizing cortical laminar patterns in the living human brain using advanced MRI methods^{11,37}. These efforts utilize different contrast mechanisms (T_1 , T_2 , T_2^* , phase, susceptibility, magnetization transfer, and dMRI) to study cortical gray matter organization on the laminar level³⁸⁻⁴². Among these methods, T_1 -weighted approaches have proven to be the most studied and demonstrated to be robust and practical, with early studies showing that myelination shortens T_1 values and that T_1 -weighted images contain laminar signatures reflecting the underlying myeloarchitecture, revealing six T_1 -defined components that correspond to histological layers⁴³⁻⁴⁵. Subsequent studies extended this framework by leveraging quantitative T_1 inversion-recovery MRI protocols for whole-brain laminar mapping while addressing partial volume effects^{46,47}, modeling cortical laminar connectivity⁴⁸⁻⁵⁰, and applying laminar imaging in clinical populations, such as detecting dyslamination in epilepsy⁵¹. Although T_1 -derived laminar patterns have recently been shown to correlate with cytoarchitectonic regions⁵², T_1 relaxation primarily reflects myeloarchitecture rather than cytoarchitecture and therefore provides an indirect measure of cellular organization¹¹.

(P 6, Line 114) Beyond these T_1 -based approaches, dMRI provides a distinct and complementary source of microstructural contrast that is sensitive to complex features such as fiber orientation, cellular density, and other microscopic tissue properties³⁴⁻³⁶. Increasing evidence suggests that the cortical diffusion signal is layer-dependent and reveals microstructural features that are complementary to conventional T_1 -weighted or quantitative relaxometry methods^{53,54}. Even basic diffusion tensor imaging (DTI) shows depth-dependent variations in fractional anisotropy and radial diffusion within the cortex^{53,55,56}, while

the fiber orientation distribution function (fODF) exhibits robust radial and tangential patterns that vary with cortical curvature and laminar architecture^{53,56,57}. These findings underscore the strong potential of dMRI for providing a more comprehensive characterization of cortical cyto- and myeloarchitecture *in vivo*. Such laminar sensitivity can be further supported by advanced biophysical modeling that disentangles the heterogeneous cellular contributors to the diffusion signal and improves specificity to compartment-level features reflecting the underlying laminar organization³⁵.

(P 8, Line 153) Soma and neurite compartments reflect distinct biological properties related to cortical architecture, where intra-neurite fraction is sensitive to axonal and dendritic organization associated with myeloarchitecture, complementing established myelin-sensitive contrasts. The intra-soma fraction provides additional sensitivity to cell body density and captures cytoarchitectonic features, thereby offering clear advantages over existing dMRI contrasts^{76,79}.

(P 23, Line 446) Since cortical fODFs exhibit distinct radial and tangential organization across laminae⁵³, integrating fODF-based orientation information with SANDI-derived compartment metrics may enhance the laminar sensitivity of dMRI-based cortical characterization. In parallel, advances in MRI-based layer segmentation are expected to strengthen further the translational potential of the laminar SANDI framework^{11,37}. Incorporating layer-specific T_1 -based segmentation could enable more accurate assignment of SANDI-derived microstructural metrics to their corresponding cortical layers⁴⁶ and provide a more direct *in vivo* surrogate for cytoarchitectonic and myeloarchitectonic organization.

Referee : 3

This study explores *in vivo* the cyto- and myeloarchitectonic organization of the human cerebral cortex using the Soma and Neurite Density Imaging (SANDI) model and high-gradient diffusion MRI (dMRI) (Connectome 2.0 scanner with 500 mT/m gradient strength), enabling unprecedented *in vivo* microstructural imaging. Compared to the

earlier Connectome 1.0 (300 mT/m), Connectome 2.0 offers higher resolution, shorter diffusion times, and improved sensitivity to cellular components (e.g., somas, neurites).

Cortical depth-dependent analysis of SANDI metrics provided a detailed characterization of cortical cytoarchitecture and myeloarchitecture in the living human brain, revealing distinct profiles that align with established histological patterns, particularly in the cell body density distribution and myelinated fibres across cortical layers. The results also suggest that Connectome 2.0 enhances microstructural specificity, particularly in detecting neurite density.

This work bridges postmortem histology and in vivo imaging, advancing non-invasive cortical microstructure characterization; it represents a significant methodological advancement in in vivo cortical microstructure imaging, with implications for basic neuroscience and clinical research.

The manuscript is very well written and contains all the relevant methodological details. The results are summarized and presented in a very informative way and the analysis is rigorous. I have only a few recommendations and comments for the authors:

Overall response: We thank the reviewer for the positive comments and recognition of the contribution of this work to the field. The reviewer's feedback has helped us improve the super-resolution image processing description, clarify the clinical translation of the SANDI framework and importance for basic neuroscience, and clarify the code availability for reproducibility. Below, we address the detailed comments in-depth:

[Referee 3 Comment 1] In the first paragraph of the Results, as well as in Figure 1; please mention and show schematically that dMRI data were super-resolved from 2 mm³ to 1mm³ isotropic

resolution.

Figure 2. Framework for cortical depth-dependent microstructure analysis of SANDI metrics, with comparisons to histological atlases.

Response: We agree that it is important to clearly indicate the use of super-resolution processing to enhance spatial specificity in our cortical analyses. In response, we have updated the first paragraph of the Results section and revised **Figure 2** to explicitly state that the diffusion MRI data were super-resolved from a native resolution of 2 mm³ to 1 mm³ isotropic resolution prior to SANDI model fitting.

Updates in the revised manuscript

(P 10, Line 187) To assess cortical microstructural features in the living human brain and to relate these depth-dependent patterns to cytoarchitectonic and myeloarchitectonic histological atlases, we applied a framework using a multi-step analysis pipeline (**Figure 2**) that includes anatomical parcellation, surface reconstruction, super-resolution image processing of the diffusion-weighted images (2 mm to 1 mm isotropic; **Supplementary Figure S1**) prior to SANDI fitting, deep learning-based supragranular/infragranular segmentation, and laminar sampling of SANDI metrics (intra-soma signal fraction f_{is} and intra-neurite signal fraction f_{in}) using FreeSurfer’s “mri_vol2surf”.

[Referee 3 Comment 2] Regarding the method for the super-resolution from Coupe et al NeuroImage 2013; it is unclear to me from the cited works and the results shown in the Supplementary Materials how accurate is the super resolution of high diffusion-weighted MRI images and how confident in the model prediction one can be (e.g., exclude hallucinations)? All the experiments and quantitative evaluations published are performed at relatively low b values and using simple signal representations (e.g., DTI). The authors should provide quantitative evidence of unbiased dMRI signal prediction and corresponding SANDI metrics when applying super-resolution to high b values. For example, one experiment could be to artificially down-sample the 2 mm³ dMRI to 4 mm³ ; add some extra Rician noise to match the original 2 mm³ SNR, and then apply the super-resolution method to predict the higher resolution, 2 mm³ dMRI. The existing ground truth at 2 mm³ can then be used to estimate, especially for the cortical voxels, mean squared error, mean absolute error and SSIM between predicted dMRI signal and SANDI maps and the ground truth ones.

Supplementary Figure S3. Super-resolution image processing for SANDI-derived intra-soma signal fraction f_{is} maps. **Top:** Intra-soma signal fraction f_{is} at 2 mm isotropic resolution and the corresponding 1 mm isotropic images obtained using different upsampling methods (nearest-neighbor interpolation, cubic interpolation, and super-resolution). **Bottom:** Quantitative validation comparing the ground-truth 2 mm image with a 4 mm down-sampled version that was subsequently upsampled back to 2 mm using the same methods.

Response: We thank the reviewer for this valuable comment. We fully agree that quantitative validation of the super-resolution performance is essential, particularly at high b values. Following the reviewer's suggestion, we performed an additional comparison using the SANDI-derived intra-soma signal fraction f_{is} maps to assess super-resolution accuracy compared to conventional interpolation methods such as nearest-neighbor and cubic interpolation. The super-resolved image at 1 mm isotropic resolution preserved fine cortical details and exhibited sharper gray-white matter contrast compared to

conventional interpolation as shown in the **Supplementary Figure S3 (top)**.

For the further quantitative evaluation, the original 2 mm isotropic diffusion MRI data were artificially downsampled to 4 mm isotropic resolution. The down-sampled data were then up-sampled back to 2 mm using three different methods (nearest-neighbor interpolation, cubic interpolation, and our super-resolution approach), and the f_{is} maps were compared with the 2 mm ground truth. As shown in the **Supplementary Figure S3 (bottom)**, the super-resolved images achieved the lowest mean squared error (MSE) and mean absolute error (MAE) values and the highest structural similarity index measure (SSIM) relative to the ground truth.

Supplementary Figure S2. SANDI-derived microstructural metrics maps at original resolution (left, 2 mm isotropic) and super-resolution (right, 1 mm isotropic). The super-resolution images are derived from self-similarity-based super-resolution image processing.

In addition, we have included representative examples of all SANDI-derived metric maps reconstructed at 1 mm isotropic resolution in **Supplementary Figure S2**. These results demonstrate that the super-resolution approach consistently enhances the visualization of microstructural properties across all SANDI-derived metrics. Methodological details and additional figures are provided in the

Supplementary Note 1.

Updates in the revised manuscript

(P 11, Line 209) The high-resolution (1 mm isotropic) dMRI-derived SANDI metrics generated using the super-resolution technique⁸²⁻⁸⁴ improved the visualization of detailed microstructural features and effectively reduced partial volume effects compared to the lower-resolution data (**Supplementary Figure S2-S3**; see **Supplementary Note 1** for details on the super-resolution imaging processing technique).

(P 23, Line 452) **Improvement of super-resolution reconstruction and high-resolution sub-millimeter imaging techniques will be essential for improving the sensitivity of laminar sampling and minimizing partial volume effects.**

[Referee 3 Comment 3] Regarding the results from Connectom 1.0, I have a couple of suggestions. 1) the authors could use the results on fis to argue more in favour of potential clinical translation, in the Discussion section: the dMRI protocol used on the Connectom 1.0 can be now easily mirrored on commercial clinical top-end scanners, such as the Siemens Cima X. As the clinical community will adopt these new technologies, replication studies and application to diseases will be feasible at scale. 2) The authors show that fis from Connectom 1.0 is not significantly different from Connectom 2.0, while fin it is. However, although fin estimates from Connectom 1.0 are significantly lower than those from Connectom 2.0, the ability of fin to capture myeloarchitectural characteristics across cortical layers may still be preserved. The authors should provide in supplementary Materials a similar analysis as in Figure 4, to show whether the impact of the different dMRI protocols between Connectom 1.0 and 2.0 is only a systematic shift of the values, which does not compromise differential analysis and sensitivity to myeloarchitecture.

Response: We thank the reviewer for these constructive suggestions regarding the analysis and interpretation of the Connectome 1.0 results. We have addressed the reviewer’s comments in the revised manuscript by including supplementary analyses using the Connectome 1.0 dataset and by expanding the discussion to clarify both (1) superior laminar microstructural sensitivity of Connectome 2.0 compared with Connectome 1.0 and (2) clinical translatability of the proposed laminar SANDI framework to the Cima.X, MAGNUS, and other high-performance gradient systems that are now commercially available.

Supplementary Figure S5. SANDI-derived microstructural metrics on Connectome 2.0 (C2.0) and Connectome 1.0 (C1.0) scanners. Left: Cortical maps of the intra-soma signal fraction f_{is} and intra-neurite signal fraction f_{in} derived from SANDI, averaged across 21 individuals. The third column shows the P -values of vertex-wise differences in these metrics between Connectome 2.0 and Connectome 1.0, indicating regions with statistically significant differences. **Right:** Boxplots summarizing intra-soma signal fraction f_{is} and intra-neurite signal fraction f_{in} across individuals ($N=21$ for both scanners), including a subset of participants who were scanned on both scanners ($N=3$), with statistically significant differences of f_{in} (*: $FDR-P < 0.05$) between Connectome 2.0 and Connectome 1.0 scanners. The box represents the 1.96 standard error of the mean (95% confidence interval), and the line represents the 1 standard deviation.

(1) Superior laminar microstructural sensitivity of Connectome 2.0 compared with Connectome 1.0

To clarify in what sense Connectome 2.0 is superior to Connectome 1.0, we now provide detailed quantitative evidence demonstrating its superior microstructural sensitivity. The higher intra-neurite signal fraction f_{in} values observed with the Connectome 2.0 scanner (compared with Connectome 1.0) in **Supplementary Figure S5** likely reflect the benefits of shorter diffusion time and echo time (TE), which together minimize inter-compartmental water exchange and improve the overall signal-to-noise ratio (SNR), thereby enhancing sensitivity to f_{in} . Additionally, our recent work using the Connectome 2.0 scanner has demonstrated superior sensitivity and microstructural contrast for soma-related SANDI

metrics compared with Connectome 1.0 (Ramos-Llordén et al., *Nature Biomedical Engineering*, 2025). In particular, the intra-soma signal fraction f_{in} revealed greater regional contrast across Brodmann areas (4, 3a, 3b, and 1), together with smaller inter-subject variability on Connectome 2.0, closely matching prior histological findings. These distinctions were less pronounced when estimated on Connectome 1.0. Moreover, Connectome 2.0 showed increased sensitivity to smaller soma populations. Together, these results highlight that Connectome 2.0 provides better microstructural sensitivity due to its stronger gradient strength and by enabling access to shorter diffusion times in the measurements.

(2) Clinical translatability of the proposed laminar SANDI framework

Supplementary Figure S8. SANDI-derived intra-neurite signal fraction f_{in} on the Connectome 1.0 scanner, alongside the myelin staining data from the myeloarchitecture atlas across the cortical depths. In the myelin staining data, darker colors (i.e., lower intensity values) correspond to higher myelin concentration across regions defined by Nieuwenhuys’ parcellation. The data of myelin staining intensity across cortical depths and the figure of histology are sourced from Foit et al., 2022⁵, with permission from Elsevier.

To further address the reviewer’s comment on the relevance of the Connectome 1.0 results, we also discussed that, despite its lower absolute sensitivity, Connectome 1.0 still preserved the ability to capture laminar microstructural organization. As shown in the newly added **Supplementary Figure S8**, the depth-dependent profiles of f_{in} measured on Connectome 1.0 closely follow those obtained on Connectome 2.0. This supports the feasibility of laminar SANDI measurements on high-performance clinical MRI systems, including the Siemens Cima.X and GE MAGNUS scanners, and potentially more broadly on currently available and future 3T clinical scanners, thereby facilitating the translational potential of the proposed framework.

Updates in the revised manuscript

(P 14, Line 271) The Connectome 1.0 data preserved the depth-dependent f_{in} profile, showing an increase toward the white matter with overall lower values and a correlation with myeloarchitecture patterns ($r=-0.18$; $P=0.002$) comparable to that observed on Connectome 2.0 (Supplementary Figure S8).

(P 19, Line 353) **Enhanced microstructural sensitivity with Connectome 2.0**

The Connectome 2.0 scanner shows enhanced microstructural sensitivity relative to Connectome 1.0, enabling more precise characterization of both neurite and soma compartments. The stronger gradient system allows for substantially shorter diffusion times and echo times, which reduce inter-compartmental water exchange effects^{74,92-94} that are not modeled in SANDI⁷⁸ and improve SNR, resulting in higher intra-neurite signal fraction f_{in} values. Beyond neurites, although whole-cortex averages of the intra-soma signal fraction f_{is} did not differ between scanners, more detailed subregional analyses using Connectome 2.0 revealed improved performance for soma-related SANDI metrics. Specifically, the intra-soma signal fraction f_{is} exhibited greater regional contrast across Brodmann areas (BA4, 3a, 3b, and 1), closely matching prior histological findings, and reduced inter-subject variability compared with Connectome 1.0³⁰. In addition, Connectome 2.0 shows greater sensitivity to smaller soma populations.

(P 23, Line 438) Importantly, our results further support this translatability. Although the Connectome 2.0 scanner provides greater sensitivity than its predecessor³⁰, the Connectome 1.0 scanner still retains sufficient sensitivity to capture depth-dependent microstructural organization. These findings support the feasibility of implementing the cortical laminar SANDI framework on high-performance clinical systems, including the Siemens 3T Cima.X and GE MAGNUS²⁸ scanners, and potentially more broadly on currently available and future 3T clinical scanners.

[Referee 3 Comment 4] While I do understand why sharing “upon request” of the MRI data may be preferred; I am not in favour of not sharing openly the codes used to produce the results. I strongly encourage the authors to revise their Data availability statement to embrace a more open and reproducible approach by sharing all the codes on an open access platform, for example GitHub.

Response: In response to the reviewer’s suggestion, we have revised the statement regarding code availability. We now provide all scripts used for data preprocessing and analysis through a public GitHub repository at <https://github.com/Connectome20/Laminar-SANDI>.

Updates in the revised manuscript

(P 32, Line 647) **The data used in the study are available upon direct request, along with the conditions for their sharing or re-use. The code for preprocessing and analysis is publicly available on GitHub at <https://github.com/Connectome20/Laminar-SANDI>.**

[Referee 3 Comment 5] The Discussion section is somewhat repetitive, particularly in reiterating key findings about Connectome 2.0's advantages, SANDI-derived metrics, and comparisons with histology. To improve readability without losing critical information, I suggest: merging scanner comparisons into a single, cohesive section; combining depth-dependent findings (e.g., layer-specific fis/fin trends) with histological validations and streamlining discussion of limitations by avoiding redundant mentions of model constraints/limitations.

Response: We appreciate the reviewer’s thoughtful feedback. In response, we have carefully revised the Discussion section to reduce redundancy and improve clarity. The first paragraph of the Discussion

section now provides a succinct summary of the main findings, and the remaining parts of the Discussion have been structured into the following sections: (1) Enhanced microstructural sensitivity with Connectome 2.0, (2) Laminar and regional variations in cortical microstructure with histological correspondence, (3) Cortical geometry (curvature) and laminar organization, (4) Clinical translation & future scientific directions, and (5) Limitations. We believe this reorganization improves the flow and readability of the Discussion.

Updates in the revised manuscript

(P 19, Line 344) Collectively, our findings show the promise of using high-performance gradient dMRI for detailed mapping of cortical microstructure *in vivo*, bridging the gap between traditional histology and *in vivo* neuroimaging, with a wide range of potential applications in neuroscience, neurology, and psychiatry. For example, the approaches used here may be used to detect subtle alterations in cellular architecture within and across cortical layers and columns associated with aging and neurodegenerative diseases. More fundamentally, if architectonic patterning of the cortex predicts connectivity^{86,88-91}, then the ability to map tissue microstructure across cortical layers may enable the development of microstructure profile-based connectomes and advance an understanding of how the human brain is wired across scales⁸⁸.